# Partially Observable Reinforcement Learning with Memory Traces

**Onno Eberhard** [1 2]  **Michael Muehlebach** [1]  **Claire Vernade** [2]

## Abstract

Partially observable environments present a considerable computational challenge in reinforcement learning due to the need to consider long histories. Learning with a finite window of observations quickly becomes intractable as the window length grows. In this work, we introduce *memory traces*. Inspired by eligibility traces, these are compact representations of the history of observations in the form of exponential moving averages. We prove sample complexity bounds for the problem of offline on-policy evaluation that quantify the return errors achieved with memory traces for the class of Lipschitz continuous value estimates. We establish a close connection to the window approach, and demonstrate that, in certain environments, learning with memory traces is significantly more sample efficient. Finally, we underline the effectiveness of memory traces empirically in online reinforcement learning experiments for both value prediction and control.

## 1. Introduction

Learning and acting in partially observable environments requires memory. It is generally not enough to act simply according to the most recent observation, and an agent must therefore keep track of the past, as every piece of information is likely to be relevant for future actions. In the absence of additional assumptions, optimal behavior requires the agent to reason over its entire history of observations. This space quickly becomes intractable, and just like with continuous state spaces in fully observed systems, it is therefore necessary to resort to approximation. This article investigates the fundamental limitations and trade-offs arising from these approximations. We focus on extracting features from the history of observations and analyze the complexity of learning functions of these features. In contrast to meth-

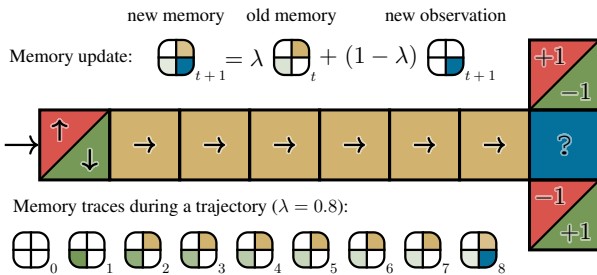

*Figure 1.* Illustration of the memory trace mechanism in the *T-maze* environment. The agent starts in the leftmost tile and receives an observation that reveals if the reward at the end of the corridor is at the top or bottom. The agent has to remember this information until the last step. It can be seen that this information has faded in $z_8$ (the final memory trace), but has not completely disappeared.

ods based on recurrent neural networks that learn directly from the stream of observations (e.g., Ni et al., 2022), this feature-based approach provides an analysis framework that is mathematically tractable.

The archetypical feature in partially observable reinforcement learning (RL) is the *length-$m$ window*, which truncates the history and only keeps the $m$ most recent observations. In deep RL, this approach is known as *frame stacking* (Mnih et al., 2015). Under certain assumptions, such as observability (Golowich et al., 2023; Liu et al., 2022) or multi-step decodability (Efroni et al., 2022), it can be shown that such a window is sufficient for good behavior. However, in many settings, the window would have to be very long to be useful, which is problematic, as the complexity of learning general functions of length-$m$ windows scales exponentially in $m$. To address this issue, this paper introduces a different feature that we call the *memory trace*. This feature is inspired by eligibility traces, and consists of an exponential moving average of the stream of observations. Thus, upon receiving a new observation $y_t$, the memory trace $z_t$ is moved closer toward $y_t$:

$$z_t = \lambda z_{t-1} + (1 - \lambda)y_t, \tag{1}$$

where $\lambda \in [0, 1)$ is a forgetting factor. This mechanism is illustrated in Fig. 1 in the *T-maze* environment (Bakker, 2001), where we took inspiration from Allen et al. (2024).

Our results are concerned with the problem of *offline on-policy evaluation*, which is an ideal setting for studying

---

[1]Max Planck Institute for Intelligent Systems, Tübingen, Germany [2]University of Tübingen. Correspondence to: Onno Eberhard <oeberhard@tue.mpg.de>.

*Proceedings of the 42nd International Conference on Machine Learning*, Vancouver, Canada. PMLR 267, 2025. Copyright 2025 by the author(s).

the window and memory trace features in terms of sample efficiency. We find that learning with windows is equivalent, in terms of capacity and sample complexity, to learning Lipschitz continuous functions of memory traces *if and only if* $\lambda < \frac{1}{2}$. For larger $\lambda$, we demonstrate that there are environments where learning Lipschitz functions of memory traces is significantly more efficient than learning with windows to achieve a desired return error. In particular, we show that the T-maze (Fig. 1) is such an environment and also present an illustrative two-state environment to provide intuition.

Turning to *online* reinforcement learning, we show that memory traces are easily incorporated into existing algorithms to provide a scalable alternative to windows. We empirically demonstrate that, under temporal difference learning with linear function approximation, memory traces significantly outperform the general length-$m$ window approach in a simple random walk experiment. Finally, we show that memory traces can handle considerably longer memory requirements than the frame stacking approach by evaluating a proximal policy optimization (PPO; Schulman et al., 2017) agent in the T-maze environment.

## 2. Related work

The topic of memory has a long-standing history in reinforcement learning. Over the years, there have been many proposals for how to design effective memory, from *utile distinction memory* (McCallum, 1993), which, recognizing that not all length-$m$ histories are important, builds a tree of histories that are useful for value prediction, to *neural Turing machines* (Graves et al., 2014), in which a deep learning agent is equipped with external memory to write to and read from. Common types of memory for deep reinforcement learning include recurrent neural networks (e.g., Hausknecht & Stone, 2015) and transformer architectures (e.g., Esslinger et al., 2022). While some of these designs have lead to empirical success (e.g., Vinyals et al., 2019), the only type of memory that is theoretically well understood is the length-$m$ window, which has been studied extensively (Efroni et al., 2022; Golowich et al., 2023; Cayci et al., 2024).

The memory trace that we present in this work is inspired by the *eligibility trace* (e.g., Sutton & Barto, 2018, Chapter 12), which can also be interpreted as a type of memory. The relevance of eligibility traces for learning in partially observable environments has been studied before (Loch & Singh, 1998; Allen et al., 2024), but they have, to the best of our knowledge, not been analyzed as a memory for RL.

## 3. Preliminaries

**POMDPs.** We consider the problems of prediction and control in a finite partially observable Markov decision process (POMDP). The state space is $\mathcal{X} = \{x^1, \ldots, x^{|\mathcal{X}|}\}$, the

action space is $\mathcal{U} = \{u^1, \ldots, u^{|\mathcal{U}|}\}$, and the observation space is $\mathcal{Y} = \{y^1, \ldots, y^{|\mathcal{Y}|}\} \subset \mathcal{Z}$, where $\mathcal{Z}$ is a Euclidean space. For example, if $\mathcal{Y}$ is one-hot, then $y^i$ is the $i^{\text{th}}$ standard basis vector of $\mathbb{R}^{|\mathcal{Y}|} \doteq \mathcal{Z}$. The POMDP is described by the transition dynamics $p(x_{t+1} \mid x_t, u_t)$, the emission probabilities $p(y_t \mid x_t)$, and the initial state distribution $p(x_0)$. The reward function $r : \mathcal{Y} \to [\underline{r}, \bar{r}]$ maps observations to rewards.[1] If a policy is fixed, then the POMDP reduces to a hidden Markov model (HMM).

**Memory traces and windows.** A HMM's *history* at time $t$ is the sequence of observations up to that time: $h_t \doteq (y_t, y_{t-1}, \ldots)$. It is often convenient to let the time $t = 0$ stand for the *current* time step and write $h \doteq h_0$ and $y \doteq y_0$. A history $h \in \mathcal{Y}^{|h|}$ may be of finite or infinite length $|h| \in \{0, 1, \ldots, \infty\}$. The *length-$m$ window* is obtained from a history by truncation:

$$\text{win}_m(h) \doteq (y_0, y_{-1}, \ldots, y_{-m+1}).$$

The *memory trace* corresponding to a history $h$ is defined as

$$z_\lambda(h) \doteq (1 - \lambda) \sum_{k=0}^{|h|-1} \lambda^k y_{-k}. \qquad (2)$$

Given a history $h$, we often write $z_t \doteq z_\lambda(h_t)$, and define $z \doteq z_0$. We can then rewrite (2) recursively as to obtain (1) for $t \in \{0, \ldots, -|h| + 1\}$, with $z_{-|h|} \doteq 0$. The set of all length-$m$ memory traces is defined as

$$\mathcal{Z}_\lambda^m \doteq \{z_\lambda(h) \mid h \in \mathcal{Y}^m\},$$

where $m \in \{0, 1, \ldots, \infty\}$, and we define $\mathcal{Z}_\lambda \doteq \mathcal{Z}_\lambda^\infty$. Both windows and memory traces can be extended to include past actions in addition to observations.

**Covering numbers.** Many of our results are based on the concept of *covering numbers*. Let $(X, \rho)$ be a metric space and $T \subset X$. A finite set $S \subset X$ is said to $\epsilon$-*cover* $T$, for some $\epsilon > 0$, if for every $x \in T$, there exists a $y \in S$ such that $\rho(x, y) \leq \epsilon$. The $\epsilon$-*covering number* $N_\epsilon(T)$ is the cardinality of the smallest $\epsilon$-cover of $T$. The *metric entropy* of $T$ is defined as $H_\epsilon(T) \doteq \log N_\epsilon(T)$, and the *Minkowski dimension* of $T$ is defined as $\dim(T) \doteq \lim_{\epsilon \to 0} \frac{H_\epsilon(T)}{\log 1/\epsilon}$, assuming the limit exists. We deal with both Euclidean spaces, in which case $\rho$ is the Euclidean norm, and functional spaces, where $\rho$ is the sup-norm $\|f\|_\infty \doteq \sup_x |f(x)|$.

**Offline on-policy evaluation.** We study the problem of *offline on-policy evaluation*. This setting assumes the availability of a dataset $\mathcal{D} = \{\tau_i\}_{i=1}^n$ consisting of $n$ trajectories

$$\tau_i = (\ldots, y_{-2}, y_{-1}, y_0, y_1, y_2, \ldots)_i \sim \mathcal{E}$$

---

[1]Sometimes $r$ is defined to map from $\mathcal{X} \times \mathcal{U}$. Our definition is equivalent (since it is always possible to extend $\mathcal{Y}$ to include rewards) and emphasizes that rewards are observed quantities.

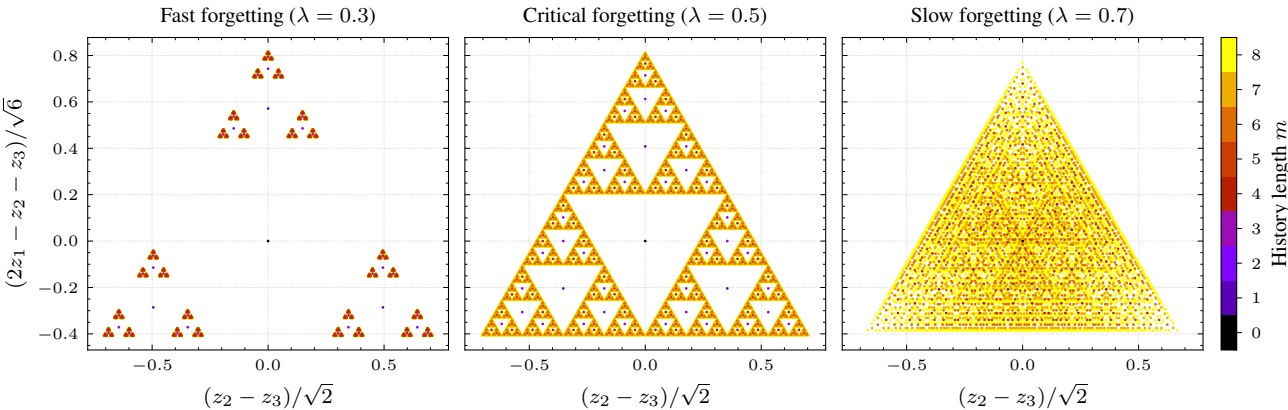

*Figure 2.* A visualization of *trace space*, where $\mathcal{Y}$ is one-hot with $|\mathcal{Y}| = 3$. The memory traces form Sierpiński triangle patterns, with $\lambda$ controlling the dilation between triangles. The center point of each triangle corresponds to the length-$m$ window shared by all traces that make up the triangle. A similar visualization of the trace space for one-hot observations with $|\mathcal{Y}| = 4$ is shown in Fig. 6.

that are drawn independently from an environment $\mathcal{E}$ (a hidden Markov model with a reward function $r$). We assume that all trajectories are of infinite length in both positive and negative time, and consider the problem of approximating the value function $v : \mathcal{Y}^\infty \to [\underline{v}, \bar{v}]$,

$$v(h) \doteq \mathbb{E}_\mathcal{E}\left[\sum_{t=0}^\infty \gamma^t r(y_{t+1}) \mid h_0 = h\right],$$

where $\underline{v} \doteq \underline{r}/(1 - \gamma)$, $\bar{v} \doteq \bar{r}/(1 - \gamma)$, and $\gamma \in [0, 1)$ is a discount factor. Given a function class $\mathcal{F} \subset \{f : \mathcal{Y}^\infty \to [\underline{v}, \bar{v}]\}$, our goal is to find the function $f \in \mathcal{F}$ that minimizes the *return error*

$$\mathcal{R}_\mathcal{E}(f) \doteq \frac{1}{2}\mathbb{E}_\mathcal{E}\left[\left\{f(y_0, y_{-1}, \dots) - \sum_{t=0}^\infty \gamma^t r(y_{t+1})\right\}^2\right].$$

As this expectation cannot be computed directly, we instead consider functions that minimize the *empirical return error*

$$\mathcal{R}_n(f) \doteq \frac{1}{2n}\sum_{\tau \in \mathcal{D}}\left\{f(y_0, y_{-1}, \dots) - \sum_{t=0}^\infty \gamma^t r(y_{t+1})\right\}^2,$$

which we analyze via generalization bounds.

## 4. The geometry of trace space

Perhaps surprisingly, it turns out that, under mild conditions, the memory trace preserves all information of the history.

**Theorem 4.1** (Finite injectivity). *If $\lambda \in (0, 1) \cap \mathbb{Q}$ and $\mathcal{Y}$ is linearly independent, then the memory trace is injective: if $h$ and $\bar{h}$ are distinct finite histories, then $z_\lambda(h) \neq z_\lambda(\bar{h})$.*

*Proof.* Although the histories $h$ and $\bar{h}$ are finite, we can 0-*pad* them by defining $y_{-k} \doteq 0$ for $k \geq |h|$ and $\bar{y}_{-k} \doteq 0$ for $k \geq |\bar{h}|$ (the memory trace is unaffected by this). The traces

$z_\lambda(h)$ and $z_\lambda(\bar{h})$ are only equal if their difference is zero:

$$\frac{z_\lambda(h) - z_\lambda(\bar{h})}{1 - \lambda} = \sum_{k=0}^\infty \lambda^k (y_{-k} - \bar{y}_{-k}) = \sum_{i=1}^{|\mathcal{Y}|} \alpha_i(\lambda) y^i,$$

where we have defined, using the Iverson bracket $[\cdot]$,

$$\alpha_i(\lambda) \doteq \sum_{k=0}^\infty \lambda^k \left([y_{-k} = y^i] - [\bar{y}_{-k} = y^i]\right). \tag{3}$$

As $\mathcal{Y}$ is linearly independent, the difference $z_\lambda(h) - z_\lambda(\bar{h})$ is only zero if $\alpha_i(\lambda) = 0$ for all $i$. By assumption, the histories are distinct, and therefore there must be at least one $\alpha_i$ with at least one nonzero coefficient. By (3), the highest-order coefficient $c$ of this $\alpha_i$ (which exists because the histories are finite) is thus either $+1$ or $-1$. We will now show that if $\lambda \in \mathbb{Q} \setminus \mathbb{Z}$, then $\alpha_i(\lambda) \neq 0$, completing the proof. Assume, for the sake of contradiction, that there exists a $\lambda = \frac{p}{q}$, with $p$ and $q$ coprime integers, such that $\alpha_i(\lambda) = 0$. Then, by the rational root theorem (e.g., Aluffi, 2021, Prop. 7.29), $q$ must divide $c$. However, as $c \in \{-1, +1\}$, it follows that $q \in \{-1, +1\}$, and so $\lambda$ is an integer. Thus, we can conclude that $\alpha_i(\lambda) \neq 0$ and hence $z_\lambda(h) \neq z_\lambda(h')$. $\square$

In Appendix A, we show with two counterexamples that injectivity is not guaranteed if $\lambda$ is irrational, or if the histories are of infinite length. We also show that the result can be extended to linearly dependent observation spaces (Theorem A.3), in which case the set of $\lambda$ that guarantee injectivity is potentially more complex, but is still dense in $(0, 1)$.

Injectivity of $z_\lambda$ implies that the complete history $h$ can be perfectly reconstructed from the memory trace vector $z_\lambda(h)$. Thus, the memory trace is an equivalent representation of the history. However, the exponential decay of (2) imparts additional structure on the space of memory traces, which

makes them attractive representations for learning. We now briefly investigate this structure, and in the following sections we analyze the efficiency of learning with memory traces compared to window-based learning.

If $\mathcal{Y}$ is one-hot, then all memory traces $z \in \mathcal{Z}_\lambda^m$ lie in a $(|\mathcal{Y}| - 1)$-dimensional affine subset of $[0, 1]^{|\mathcal{Y}|}$, as can be verified by summing over the elements of $z \in \mathcal{Z}_\lambda^m$:

$$\sum_{i=1}^{|\mathcal{Y}|} z_i = (1 - \lambda) \sum_{k=0}^{m-1} \lambda^k \sum_{i=1}^{|\mathcal{Y}|} (y_{-k})_i = 1 - \lambda^m. \quad (4)$$

In Fig. 2, the sets $\mathcal{Z}_\lambda^m$ for $|\mathcal{Y}| = 3$ are visualized by orthogonal projection onto this two-dimensional subset. It can be seen that the distance between traces greatly depends on $\lambda$. This can be quantified as follows.

**Lemma 4.2** (Concentration). *Let $h$ and $\bar{h}$ be two histories of one-hot observations such that $\text{win}_m(h) = \text{win}_m(\bar{h})$ for some $m$. Then, the corresponding traces satisfy*

$$\|z_\lambda(h) - z_\lambda(\bar{h})\| \leq \sqrt{2}\lambda^m.$$

*Proof.* See Appendix B. □

Intuitively, this results states that it is hard to distinguish between traces that only differ in observations from the far past. While Theorem 4.1 guarantees the existence of functions that map traces to arbitrarily chosen values, the concentration result shows that these functions may have very large Lipschitz constants. In particular, when two traces $z$ and $\bar{z}$ correspond to histories that share the last $m$ observations, the function $f$ needs to have a Lipschitz constant of at least $|f(z) - f(\bar{z})|/(\sqrt{2}\lambda^m)$ to map these traces to two different values $f(z)$ and $f(\bar{z})$.

Lemma 4.2 only gives an upper bound on the distance between traces. For a guarantee that a given Lipschitz constant is sufficient, we need a lower bound on this distance. This is the subject of the following result.

**Lemma 4.3** (Separation). *Let $h$ and $\bar{h}$ be two histories of one-hot observations such that $\text{win}_m(h) \neq \text{win}_m(\bar{h})$ for some $m$. Then, if $\lambda \leq \frac{1}{2}$, the corresponding traces satisfy*

$$\|z_\lambda(h) - z_\lambda(\bar{h})\| \geq \sqrt{2}(1 - 2\lambda)\lambda^{m-1}.$$

*Proof.* See Appendix B. □

This result shows that, if $\lambda < \frac{1}{2}$, then $z_\lambda$ is injective even for infinite histories and irrational $\lambda$. Figure 2 illustrates why the condition $\lambda < \frac{1}{2}$ is necessary: for larger $\lambda$, the traces move past each other and may overlap.

The set $\mathcal{Z}_\lambda$ has a fractal nature that can be described through its *Minkowski dimension*, which helps characterize the sample complexity of learning with memory traces.

**Lemma 4.4.** *If $\mathcal{Y}$ is one-hot, then the Minkowski dimension of $\mathcal{Z}_\lambda$ is, for all $\lambda < \frac{1}{2}$,*

$$\dim(\mathcal{Z}_\lambda) = \frac{\log|\mathcal{Y}|}{\log(1/\lambda)} \doteq d_\lambda.$$

*For all $\lambda \in [0, 1)$, we have $\dim(\mathcal{Z}_\lambda) \leq \min\{|\mathcal{Y}| - 1, d_\lambda\}$.*
*Proof.* See Appendix B. □

## 5. Complexity of offline on-policy evaluation

We now analyze the sample complexity of offline on-policy evaluation. Throughout this section, we assume that $\mathcal{Y}$ is one-hot. We are interested in the following two families of function classes: length-$m$ window-based functions

$$\mathcal{F}_m \doteq \{f \circ \text{win}_m \mid f : \mathcal{Y}^m \to [\underline{v}, \bar{v}]\},$$

and $L$-Lipschitz continuous functions in trace space

$$\mathcal{F}_{\lambda,L} \doteq \{f \circ z_\lambda \mid f : \mathcal{Z}_\lambda \to [\underline{v}, \bar{v}], f \text{ is } L\text{-Lipschitz}\}.$$

How well a function class $\mathcal{F}$ is suited for learning the value function in an environment $\mathcal{E}$ depends on the minimum achievable return error $\mathcal{R}_\mathcal{E}(\mathcal{F}) \doteq \inf_{f \in \mathcal{F}} \mathcal{R}_\mathcal{E}(f)$ and on the size of the function class, measured by the metric entropy $H_\epsilon(\mathcal{F})$. This is expressed by the following result.

**Theorem 5.1** (Hoeffding bound). *Given a dataset of $n$ trajectories from an environment $\mathcal{E}$, a function class $\mathcal{F}$, and some $\epsilon > 0$, let $\mathcal{F}^\epsilon$ be the smallest $\epsilon$-cover of $\mathcal{F}$ and $f_n \doteq \arg\min_{f \in \mathcal{F}^\epsilon} \mathcal{R}_n(f)$. Then, with probability at least $1 - \delta$,*

$$\mathcal{R}_\mathcal{E}(f_n) \leq \mathcal{R}_\mathcal{E}(\mathcal{F}) + \Delta^2\sqrt{\frac{H_\epsilon(\mathcal{F}) + \log\frac{2}{\delta}}{2n}} + \epsilon\Delta + \frac{\epsilon^2}{2},$$

*where we have defined $\Delta \doteq \bar{v} - \underline{v}$.*

*Proof.* This result combines a standard generalization bound from statistical learning theory that applies Hoeffding's inequality to a finite hypothesis class $\mathcal{F}^\epsilon$ with Lemma 5.2 (below). The full proof is in Appendix B. □

**Lemma 5.2.** *Let $\mathcal{E}$ be an environment, $\mathcal{F}$ a function class, and $\epsilon > 0$. If $\mathcal{G}$ is an $\epsilon$-cover of $\mathcal{F}$, then*

$$\mathcal{R}_\mathcal{E}(\mathcal{G}) \leq \mathcal{R}_\mathcal{E}(\mathcal{F}) + \epsilon\Delta + \frac{\epsilon^2}{2}.$$

*Proof.* See Appendix B. □

While the Hoeffding bound above does not guarantee that function classes with large metric entropy are less suitable for learning, it suggests that a good value function is more easily learned if $H_\epsilon(\mathcal{F})$ is small. The focus on metric entropy as a measure for the complexity of hypothesis classes is well established in statistical learning theory (Wainwright, 2019; Haussler, 1992; Allard & Bölcskei, 2024). We will now analyze the metric entropies of the classes $\mathcal{F}_m$ and $\mathcal{F}_{\lambda,L}$, and compare their return errors across different environments. We begin by computing the metric entropies.

**Lemma 5.3.** *Let $m \in \mathbb{N}_0$ be a window length. The metric entropy of $\mathcal{F}_m$ is, for all $\epsilon > 0$,*

$$H_\epsilon(\mathcal{F}_m) = |\mathcal{Y}|^m \log \left\lceil \frac{\Delta}{2\epsilon} \right\rceil.$$

*Thus, as a function of $m$, $H_\epsilon(\mathcal{F}_m) \in \Theta(|\mathcal{Y}|^m)$.*

*Proof.* See Appendix B. □

**Lemma 5.4.** *Let $\lambda \in [0,1)$ and $L > 0$ be a Lipschitz constant. The metric entropy of $\mathcal{F}_{\lambda,L}$ satisfies, for all $\epsilon > 0$,*

$$H_\epsilon(\mathcal{F}_{\lambda,L}) \leq \log \left\lceil \frac{\Delta}{\epsilon} \right\rceil |\mathcal{Y}| \left( \frac{2L}{\epsilon} \right)^{d_\lambda}, \quad and$$

$$H_\epsilon(\mathcal{F}_{\lambda,L}) \leq \log \left\lceil \frac{\Delta}{\epsilon} \right\rceil \left\lceil \frac{2L\sqrt{|\mathcal{Y}|-1}}{\epsilon} \right\rceil^{|\mathcal{Y}|-1}.$$

*Thus, as a function of $\lambda$ and $L$, the metric entropy satisfies*

$$H_\epsilon(\mathcal{F}_{\lambda,L}) \in \mathcal{O}\left( L^{\min\{d_\lambda, |\mathcal{Y}|-1\}} \right).$$

*Proof.* Let $\epsilon > 0$, $\lambda \in [0,1)$, and $L > 0$. We will construct two different $\epsilon$-covers of $\mathcal{F}_{\lambda,L}$ to establish the two upper bounds. The difference between these two is how the (infinite) set $\mathcal{Z}_\lambda$ is approximated. Let $S \subset [0,1]^{|\mathcal{Y}|}$ be a finite set that $\frac{\epsilon}{2L}$-covers $\mathcal{Z}_\lambda$ (in the Euclidean norm). Then, for each $z \in \mathcal{Z}_\lambda$, there exists a $w(z) \in S$ such that $\|z - w(z)\| \leq \frac{\epsilon}{2L}$. We will now show that the set $\mathcal{F}^\epsilon(S) \doteq \{f \circ w \circ z_\lambda \mid f : S \to \mathcal{V}_{\frac{\epsilon}{2}}\}$ $\epsilon$-covers $\mathcal{F}_{\lambda,L}$ (in the sup-norm), where $\mathcal{V}_\epsilon$ is a set of $\lceil \Delta/(2\epsilon) \rceil$ points in $[\underline{v}, \overline{v}]$ that $\epsilon$-covers this interval (see proof of Lemma 5.3). Let $f \in \mathcal{F}_{\lambda,L}$. Then, there exists an $L$-Lipschitz function $\hat{f} : \mathcal{Z}_\lambda \to [\underline{v}, \overline{v}]$ such that $f = \hat{f} \circ z_\lambda$. Kirszbraun's theorem (e.g., Federer, 2014, Theorem 2.10.43) tells us that the $L$-Lipschitz function $\hat{f}$ can be extended to an $L$-Lipschitz function $\bar{f} : [0,1]^{|\mathcal{Y}|} \to [\underline{v}, \overline{v}]$ with the property that $\bar{f}|_{\mathcal{Z}_\lambda} = \hat{f}$. For every $s \in S$, define $\bar{g}(s) \in \mathcal{V}_{\frac{\epsilon}{2}}$ such that $|\bar{g}(s) - \bar{f}(s)| \leq \frac{\epsilon}{2}$. This is possible because $\mathcal{V}_{\frac{\epsilon}{2}}$ is an $\frac{\epsilon}{2}$-cover of $[\underline{v}, \overline{v}]$. Now, define $g \in \mathcal{F}^\epsilon(S)$ as $g \doteq \bar{g} \circ w \circ z_\lambda$. Then, for all histories $h \in \mathcal{Y}^\infty$, with $z \doteq z_\lambda(h)$,

$$|f(h) - g(h)| = |\bar{f}(z) - \bar{g}(w(z))|$$
$$\leq |\bar{f}(z) - \bar{f}(w(z))| + |\bar{f}(w(z)) - \bar{g}(w(z))|$$
$$\leq L\|z - w(z)\| + \epsilon/2 \leq \epsilon.$$

Thus, the metric entropy of $\mathcal{F}_{\lambda,L}$ satisfies

$$H_\epsilon(\mathcal{F}_{\lambda,L}) \leq \log |\mathcal{F}^\epsilon(S)| = |S| \log |\mathcal{V}_{\frac{\epsilon}{2}}| = |S| \log \lceil \Delta/\epsilon \rceil. \tag{5}$$

To get the first inequality, we define the set $S_1 \subset [0,1]^{|\mathcal{Y}|}$ as the following set of length-$m$ memory traces:

$$S_1 \doteq \{z_\lambda(h) \mid h \in \mathcal{Y}^m\}, \quad \text{where} \quad m \doteq \left\lceil \frac{\log(2L/\epsilon)}{\log(1/\lambda)} \right\rceil_+,$$

with $(\cdot)_+ \doteq \max\{0, \cdot\}$. We first show that $S_1$ $\frac{\epsilon}{2L}$-covers $\mathcal{Z}_\lambda$, and then compute the cardinality of $S_1$. Let $z \in \mathcal{Z}_\lambda$

and $h \in \mathcal{Y}^\infty$ such that $z = z_\lambda(h)$ (this history exists by definition of $\mathcal{Z}_\lambda$). Let $z^m \doteq z_\lambda(\text{win}_m(h)) \in S_1$. Then,

$$\|z - z^m\| \leq (1-\lambda) \sum_{k=m}^\infty \lambda^k \underbrace{\|y_{t-k}\|}_{1}$$
$$= \lambda^m \leq \exp\left( \log \lambda \cdot \frac{\log(2L/\epsilon)}{\log(1/\lambda)} \right) = \frac{\epsilon}{2L},$$

proving that $S_1$ is an $\frac{\epsilon}{2L}$-cover. The cardinality of $S_1$ is

$$|S_1| = |\mathcal{Y}|^m$$
$$\leq \exp\left\{ \log |\mathcal{Y}| \left( \frac{\log(2L/\epsilon)}{\log(1/\lambda)} + 1 \right) \right\} = |\mathcal{Y}| \left( \frac{2L}{\epsilon} \right)^{d_\lambda}.$$

The first result then follows from (5).

For the second inequality, we construct a different set $S_2 \subset [0,1]^{|\mathcal{Y}|}$. Let $e_0 \doteq \frac{1}{\sqrt{|\mathcal{Y}|}} \mathbf{1}$, and extend $e_0$ to an orthonormal basis $e_0, e_1, \ldots, e_{|\mathcal{Y}|-1}$ of $\mathbb{R}^{|\mathcal{Y}|}$. We now define

$$S_2 \doteq \left\{ \frac{1}{\sqrt{|\mathcal{Y}|}} e_0 + \sum_{i=1}^{|\mathcal{Y}|-1} c_i e_i \mid c_1, \ldots, c_{|\mathcal{Y}|-1} \in G_\delta \right\},$$

where $\delta \doteq \frac{\epsilon}{2L\sqrt{|\mathcal{Y}|-1}}$, and where $G_\delta$ is a finite set of $\lceil 1/\delta \rceil$ points in $[-1,1]$ that $\delta$-covers this interval. Such a set $G_\delta$ exists by the argument presented in the proof of Lemma 5.3: taking $\lceil 1/\delta \rceil$ uniformly spaced points with equal distance $2\delta$ as the centers of $\delta$-balls, a volume of $\lceil 1/\delta \rceil (2\delta) \geq 2$ is $\delta$-covered, which is enough to cover $[-1, 1]$.

We now show that $S_2$ is an $\frac{\epsilon}{2L}$-cover of $\mathcal{Z}_\lambda$. Let $z \in \mathcal{Z}_\lambda$. Then, using (4), we have $\|z\| \leq \|z\|_1 = \sum_i z_i = 1$. This implies that $z^\top e_i \in [-1, 1]$ for all $i \in \{0, \ldots, |\mathcal{Y}| - 1\}$. Thus, for each $i$, there exists a point $k_i \in G_\delta$ such that $|z^\top e_i - k_i| \leq \delta$. Now, define $w \in S_2$ as

$$w \doteq \frac{1}{\sqrt{|\mathcal{Y}|}} e_0 + \sum_{i=1}^{|\mathcal{Y}|-1} k_i e_i.$$

From (4), we have $z^\top e_0 = \frac{1}{\sqrt{|\mathcal{Y}|}}$. Thus,

$$\|z - w\|^2 = \left\| \sum_{i=1}^{|\mathcal{Y}|-1} (z^\top e_i - k_i) e_i \right\|^2$$
$$= \sum_{i=1}^{|\mathcal{Y}|-1} |z^\top e_i - k_i|^2$$
$$\leq (|\mathcal{Y}| - 1)\delta^2 = \left( \frac{\epsilon}{2L} \right)^2.$$

Hence, $S_2$ $\frac{\epsilon}{2L}$-covers $\mathcal{Z}_\lambda$. The cardinality of $S_2$ is

$$|S_2| = |G_\delta|^{|\mathcal{Y}|-1} = \left\lceil \frac{2L\sqrt{|\mathcal{Y}|-1}}{\epsilon} \right\rceil^{|\mathcal{Y}|-1},$$

and the result follows from (5). □

These expressions are difficult to compare without additional context. In the following two sections we show that the efficiency of learning with windows and memory traces, according to Theorem 5.1 and our metric entropy upper bounds, is *equivalent* for $\lambda < \frac{1}{2}$ ("fast forgetting"), while memory traces can be significantly more efficient when $\lambda \geq \frac{1}{2}$ ("slow forgetting").

## 5.1. Fast forgetting: $\lambda < \frac{1}{2}$

Our first result states that, if there is a window length $m$ such that $\mathcal{F}_m$ achieves a certain return error at the cost of a certain complexity (measured by the metric entropy), then there is a $\lambda < \frac{1}{2}$ and a Lipschitz constant $L > 0$ such that $\mathcal{F}_{\lambda,L}$ achieves the same return error and the same complexity, up to constant factors. Thus, there exists no environment where windows outperform memory traces *in general*.

**Theorem 5.5** (Window to trace). *For every window length $m \in \mathbb{N}$ and every $\lambda < \frac{1}{2}$, there exists a Lipschitz constant $L(m) > 0$ such that, for every $\epsilon > 0$ and every environment $\mathcal{E}$,*

$$\mathcal{R}_{\mathcal{E}}(\mathcal{F}_{\lambda,L(m)}) \leq \mathcal{R}_{\mathcal{E}}(\mathcal{F}_m),$$

*and, taking $H_\epsilon(\mathcal{F}_{\lambda,L(m)})$ as a function of $m$,*

$$H_\epsilon(\mathcal{F}_{\lambda,L(m)}) \in \mathcal{O}(|\mathcal{Y}|^m) = \mathcal{O}(H_\epsilon(\mathcal{F}_m)).$$

*Proof.* Let $m \in \mathbb{N}$ and $\lambda \in [0, \frac{1}{2})$. Define $L(m)$ as

$$L \doteq \frac{\Delta}{\sqrt{2}(1-2\lambda)\lambda^{m-1}}.$$

We begin by showing that $\mathcal{F}_m \subset \mathcal{F}_{\lambda,L}$, implying that $\mathcal{R}_{\mathcal{E}}(\mathcal{F}_{\lambda,L}) \leq \mathcal{R}_{\mathcal{E}}(\mathcal{F}_m)$ for any environment $\mathcal{E}$. Let $f \in \mathcal{F}_m$. Since $\lambda < \frac{1}{2}$, Lemma 4.3 guarantees that $z_\lambda$ is invertible, and we can define $g : \mathcal{Z}_\lambda \to [\underline{v}, \bar{v}]$ as $g \doteq f \circ z_\lambda^{-1}$. We now show that $g$ is $L$-Lipschitz, implying that $f = g \circ z_\lambda \in \mathcal{F}_{\lambda,L}$. The Lipschitz constant of $g$ is the supremum of the Lipschitz ratio:

$$\mathrm{Lip}(g) = \sup_{\substack{z,\bar{z} \in \mathcal{Z}_\lambda \\ z \neq \bar{z}}} \frac{|g(z) - g(\bar{z})|}{\|z - \bar{z}\|}.$$

Let $z \neq \bar{z} \in \mathcal{Z}_\lambda$ and define $h \doteq z_\lambda^{-1}(z)$ and $\bar{h} \doteq z_\lambda^{-1}(\bar{z})$. If $\mathrm{win}_m(h) = \mathrm{win}_m(\bar{h})$, then $g(z) = g(\bar{z})$, making the Lipschitz ratio $0 \leq L$. Otherwise, if $\mathrm{win}_m(h) \neq \mathrm{win}_m(\bar{h})$, Lemma 4.3 guarantees that $\|z - \bar{z}\| \geq \sqrt{2}(1-2\lambda)\lambda^{m-1}$. As $|g(z) - g(\bar{z})| \leq \Delta$, we have $\mathrm{Lip}(g) \leq L$, showing that $f \in \mathcal{F}_{\lambda,L}$.

Given $\epsilon > 0$, the metric entropy satisfies (by Lemma 5.4)

$$H_\epsilon(\mathcal{F}_{\lambda,L}) \leq \log\left\lceil\frac{\Delta}{\epsilon}\right\rceil |\mathcal{Y}|\left(\frac{\sqrt{2}\Delta}{(1-2\lambda)\lambda^{m-1}\epsilon}\right)^{d_\lambda}$$

$$= \underbrace{\log\left\lceil\frac{\Delta}{\epsilon}\right\rceil\left(\frac{\sqrt{2}\Delta}{(1-2\lambda)\epsilon}\right)^{d_\lambda}}_{c} |\mathcal{Y}|\left(\frac{1}{\lambda}\right)^{(m-1)d_\lambda}$$

$$= c|\mathcal{Y}| \exp\left\{\log(1/\lambda)(m - 1)\frac{\log|\mathcal{Y}|}{\log(1/\lambda)}\right\}$$

$$= c|\mathcal{Y}|^m \in \mathcal{O}(|\mathcal{Y}|^m). \qquad \square$$

The next result is similar in spirit, as it shows that for every $\lambda$ and Lipschitz constant $L$ there exists a window length $m$ such that $\mathcal{F}_m$ has a return error comparable to $\mathcal{F}_{\lambda,L}$, but it does not go quite as far in terms of complexity. In the next section we show why: there *are* environments where memory traces are significantly more efficient than windows.

**Theorem 5.6** (Trace to window). *For every $\lambda \in [0,1)$, Lipschitz constant $L > 0$, and $\epsilon \in (0, L)$, there exists a window length $m(\lambda, L) \in \mathbb{N}_0$ such that, for every environment $\mathcal{E}$,*

$$\mathcal{R}_{\mathcal{E}}(\mathcal{F}_{m(\lambda,L)}) \leq \mathcal{R}_{\mathcal{E}}(\mathcal{F}_{\lambda,L}) + \epsilon\Delta + \frac{\epsilon^2}{2}$$

*and, taking $H_\epsilon(\mathcal{F}_{m(\lambda,L)})$ as a function of $\lambda$ and $L$,*

$$H_\epsilon(\mathcal{F}_{m(\lambda,L)}) \in \mathcal{O}(L^{d_\lambda}).$$

*Proof.* Let $\lambda \in [0,1)$, $L > 0$, and $\epsilon \in (0, L)$. Define $m(\lambda, L)$ as

$$m \doteq \left\lceil\frac{\log(L/\epsilon)}{\log(1/\lambda)}\right\rceil.$$

We begin by showing that for every $f \in \mathcal{F}_{\lambda,L}$, there exists a $g \in \mathcal{F}_m$ such that $\|f - g\|_\infty \leq \epsilon$. The return error bound then follows from Lemma 5.2. Let $f \in \mathcal{F}_{\lambda,L}$. Then, there exists an $L$-Lipschitz function $\hat{f} : \mathcal{Z}_\lambda \to [\underline{v}, \bar{v}]$ such that $f = \hat{f} \circ z_\lambda$. As in Lemma 5.4, we can use Kirszbraun's theorem to extend $\hat{f}$ to an $L$-Lipschitz function $\bar{f} : [0,1]^{|\mathcal{Y}|} \to [\underline{v}, \bar{v}]$ with the property that $\bar{f}|_{\mathcal{Z}_\lambda} = \hat{f}$. Now define $g \doteq \bar{f} \circ z_\lambda \circ \mathrm{win}_m \in \mathcal{F}_m$. Let $h \in \mathcal{Y}^\infty$ be a history, and define $z \doteq z_\lambda(h)$ and $z^m \doteq z_\lambda(\mathrm{win}_m(h))$. From the proof of Lemma 5.4, we know that $\|z - z^m\| = \lambda^m \leq \frac{\epsilon}{L}$. Thus,

$$|f(h) - g(h)| = |\bar{f}(z) - \bar{f}(z^m)| \leq L\|z - z^m\| \leq \epsilon.$$

The metric entropy of $\mathcal{F}_m$ satisfies (Lemma 5.3)

$$H_\epsilon(\mathcal{F}_m) = |\mathcal{Y}|^m \log\left\lceil\frac{\Delta}{2\epsilon}\right\rceil$$

$$\leq \exp\left\{\log|\mathcal{Y}|\left(\frac{\log(L/\epsilon)}{\log(1/\lambda)} + 1\right)\right\}\log\left\lceil\frac{\Delta}{2\epsilon}\right\rceil$$

$$= \log\left\lceil\frac{\Delta}{2\epsilon}\right\rceil|\mathcal{Y}|\left(\frac{L}{\epsilon}\right)^{d_\lambda} \in \mathcal{O}(L^{d_\lambda}). \qquad \square$$

While we also know that $H_\epsilon(\mathcal{F}_{\lambda,L}) \in \mathcal{O}(L^{d_\lambda})$, this is not a lower bound. In fact, Lemma 5.4 gives us a second upper bound, $H_\epsilon(\mathcal{F}_{\lambda,L}) \in \mathcal{O}(L^{|\mathcal{Y}|-1})$. In the fast forgetting regime ($\lambda < \frac{1}{2}$), this second bound is not relevant, as the following lemma shows.

**Lemma 5.7.** *Let* $|\mathcal{Y}| > 1$. *If* $\lambda < \frac{1}{2}$, *then* $d_\lambda < |\mathcal{Y}| - 1$.

*Proof.* See Appendix B. $\qquad\square$

Thus, if $\lambda < \frac{1}{2}$, learning with windows is equivalent to learning with memory traces, as far as our results go. However, as we show next, in the *slow forgetting* regime ($\lambda \geq \frac{1}{2}$), memory traces can be significantly more efficient.

### 5.2. Slow forgetting: $\lambda \geq \frac{1}{2}$

When $\lambda \geq \frac{1}{2}$, we lose the guarantee of separation (Lemma 4.3), so memory traces can be arbitrarily close together. However, not all histories are necessarily relevant for accurately representing a value function. The following result constructs an environment where most histories are irrelevant. A large $\lambda$ can push the traces that do matter apart (see Lemma 4.2), such that a smaller Lipschitz constant suffices for estimating the value function.

**Theorem 5.8** (T-maze)**.** *There exists a sequence* $(\mathcal{E}_k)$ *of environments (with constant observation space* $\mathcal{Y}$*) with the property that, for every* $\epsilon > 0$,

$$\min_{m \in \mathbb{N}} \{H_\epsilon(\mathcal{F}_m) \mid \mathcal{R}_{\mathcal{E}_k}(\mathcal{F}_m) = 0\} \in \Omega(|\mathcal{Y}|^k), \text{ and}$$

$$\min_{\lambda \in [0,1)} \min_{L \geq 0} \{H_\epsilon(\mathcal{F}_{\lambda,L}) \mid \mathcal{R}_{\mathcal{E}_k}(\mathcal{F}_{\lambda,L}) = 0\} \in \mathcal{O}(k^{|\mathcal{Y}|-1}).$$

*Proof.* The *T-maze* environment with corridor length $k$ (see Fig. 1) provides such a sequence. As the first tile has to be remembered until the end of the corridor, the window size must be at least $m = k$, which yields the first line (using Lemma 5.3). Setting $\lambda = \frac{k-1}{k}$, we show that a Lipschitz constant of $L = \sqrt{2}ek$ suffices to solve this task with zero return error. The second line then follows from Lemma 5.4. The complete proof is deferred to Appendix B. $\qquad\square$

In the T-maze example, most histories are not relevant, so their value estimates do not contribute to the return error. This makes it possible for a large $\lambda$ to reduce the necessary Lipschitz constant. Another scenario where a large $\lambda$ can be effective is a highly *stochastic* environment. Consider the following simple HMM with state space $\mathcal{X} = \{0, 1\}$ and one-hot observation space $\mathcal{Y} = \{0, 1\}$. The transition and emission probabilities are given in the diagram below.

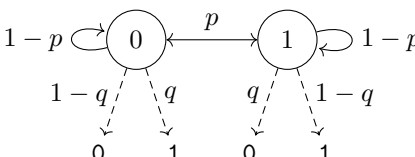

The probability of transitioning is $p$, and the probability of an 'error' in the emission (i.e., $y_t \neq x_t$) is $q$. These parameters influence how much each observation can be trusted. For example, if $q$ is large, then relying only on the most

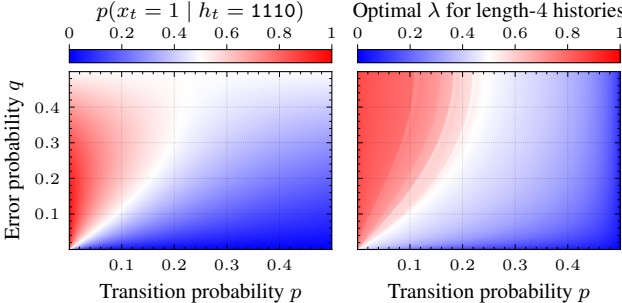

*Figure 3.* Investigating a simple 2-state HMM. (Left) The most recent observation is 0, but depending on the parameters $p$ and $q$ of the HMM, the predicted state may be either 0 or 1. (Right) The choice of $\lambda$ depends on the properties of the HMM.

recent observation may be misleading. Instead, a more reliable estimate of the state can be obtained by accumulating several observations. On the other hand, if $p$ is large, then one should not rely too much on older observations, as these are likely outdated. In the left of Fig. 3, we show how the state estimate changes based on these parameters.

Given a reward function $r$, we would like to approximate the value function $v(h)$ with a Lipschitz function $f \in \mathcal{F}_{\lambda,L}$. The optimal choice for $\lambda$ may be defined as

$$\lambda^\star \doteq \underset{\lambda \in [0,1)}{\arg\min} \inf \{L \geq 0 \mid \mathcal{R}_\mathcal{E}(\mathcal{F}_{\lambda,L}) = 0\},$$

where, for this simple environment, the choice of $r$ does not influence the result. In the right of Fig. 3, we numerically approximate $\lambda^\star$ while restricting the computation of the return error to only consider length-4 histories. We see that both plots look very similar: wherever past observations are highly informative about the the present state (the red region in the left plot), we should prefer $\lambda \geq \frac{1}{2}$. This leads to the memory trace effectively computing an average of recent observations, which is exactly the "accumulating" behavior that is needed when $q$ is large and $p$ is small.

## 6. Experiments

We now evaluate the effectiveness of memory traces as a practical alternative to windows in online reinforcement learning. Our first experiment considers the setting of online policy evaluation by temporal difference learning with linear function approximation. In our second experiment, we test the potential of memory traces for deep reinforcement learning.

**Temporal difference (TD) learning.** The environment that forms the basis for our first experiment is a modified version of *Sutton's random walk* (Sutton & Barto, 2018, Example 9.1). There are 1001 states arranged in a line with state 0 at the left and state 1000 at the right. The agent

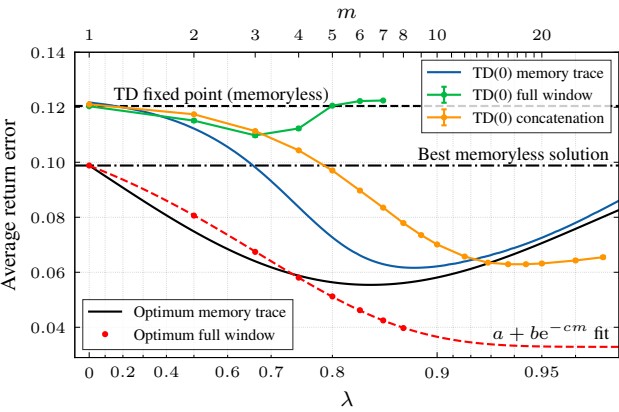

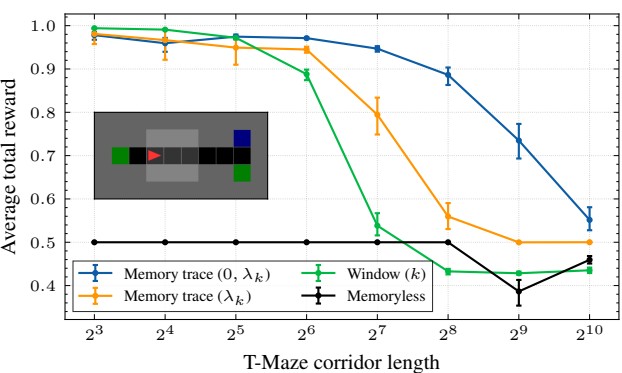

*Figure 4.* Return errors achieved with memory traces and windows in Sutton's noisy random walk environment. The experiments were repeated with 100 different random seeds and the plot contains (invisible) 95% confidence intervals for the average return error.

*Figure 5.* Average success rate of PPO in the T-maze. If a long memory is required, frame stacking is not a viable solution, and is outperformed by memory traces. The experiments were repeated with 50 random seeds and we show 95% confidence intervals.[3]

starts in the center (state $x = 500$), and in each step is randomly transported to one of the 100 states to its left or right. Should the agent be close to the edge (i.e., $x < 100$ or $x > 900$), then the excess probability mass is moved to state 500, so that going over the edge transports the agent back to the center. Additionally, going over the edge on the right gives a reward of $+1$, while going over the edge on the left gives a reward of $-1$. All other transitions give a reward of 0. This environment is partially observable: even though there are 1001 states, there are only $|\mathcal{Y}| = 11$ different observations. If the agent is in state $x$, then corresponding "bracket" is $\lfloor \frac{11}{1001} x \rfloor$. With probability 0.5, the agent observes this bracket, otherwise the agent receives a random observation between 0 and 10.

In Fig. 4, we plot the return errors achieved by TD learning when using either memory traces with linear function approximation or a length-$m$ window ("full window"). For comparison, we also show the best possible return error achievable with either type of memory. It can be seen that the trace approach significantly outperforms the window. While the window can perform well in theory, in practice, the combinatorial complexity due to the number of possible length-$m$ windows quickly becomes prohibitive. The learned weight vector has $|\mathcal{Y}|$ parameters in the memory trace approach, but $|\mathcal{Y}|^m$ parameters in the window approach. We also show the performance of an alternative window memory in which the $m$ most recent observations are concatenated into a single vector, so that the weight vector has $m|\mathcal{Y}|$ parameters. Despite this increased flexibility, we see that it does not improve performance over the exponential weighting of memory traces.

**Proximal policy optimization (PPO).** We now test the capabilities of memory traces for control. The T-maze has already been introduced (see Fig. 1). We construct a Minigrid

(Chevalier-Boisvert et al., 2024) version of this environment, shown inset in Fig. 5 (with corridor length $k = 8$), and evaluate the performance of PPO (Schulman et al., 2017). The agent starts in a colored tile on the left, and, at the end of the corridor, has to decide whether to go up or down. If the color of the terminal tile matches the starting tile, a reward of $+1$ is given. All other transitions result in a reward of 0.

In Fig. 5, we compare the average success rate per learning episode of PPO when using either memory. In this context, the window memory approach is also called *frame stacking*. To make the maze solvable, we use a window length that is equal to the corridor length $k$. For the memory trace approach, we used two parallel traces with different values of $\lambda$ as input. The first is $\lambda = 0$, corresponding to the current observation, and the second is $\lambda = \lambda_k \doteq \frac{k-1}{k}$, as derived in the proof of Theorem 5.8. Also shown are the performances of PPO when using either of these values individually (in this case, $\lambda = 0$ is a memoryless policy). We can see that the memory trace can function as a good alternative to frame stacking in tasks where a long memory is required. While frame stacking avoids the combinatorial explosion experienced in the tabular case, the neural networks still become excessively large for very long windows, and learning becomes difficult. Additional details regarding implementation and hyperparameters for both PPO and TD learning can be found in Appendix C.

## 7. Discussion & conclusion

We introduced the *memory trace*, a new type of memory for reinforcement learning. Our analysis compares this

---

[3]An average success rate of less than 0.5 is a result of premature termination with a reward of 0 if the maximum episode length of $5(k+2)^2$ steps is exceeded (where $k$ is the corridor length).

feature to the common *window* memory, where we find that the two concepts are closely linked and even equivalent in some cases. However, we also demonstrate that there are environments that can be efficiently solved with memory traces but not with windows, by characterizing the complexity of learning Lipschitz functions. In contrast to this, we show that the converse statement is not true; there is no environment that can be efficiently solved with windows, but not with memory traces. Our focus on Lipschitz functions is motivated by the geometry of the space of memory traces, and is also of practical relevance, as neural networks have been shown to learn functions with lower Lipschitz constants more easily (Rahaman et al., 2019). We further demonstrate in experiments that memory traces provide an effective alternative to windows.

Theoretical efforts in the community have mainly focused on analyzing window memory, despite its poor scaling properties. Unfortunately, the resulting guarantees are based on observability assumptions that are often violated (e.g., by overcomplete environments such as Sutton's noisy random walk). Memory traces, in contrast, provide a practical feature that is mathematically tractable, and our analysis is a first step towards understanding such memory features. An important extension that we leave to future work is a characterization of the solvability of environments with memory traces and Lipschitz continuous functions in terms of explicit properties of the transition and emission probabilities.

## Acknowledgments

We thank the International Max Planck Research School for Intelligent Systems (IMPRS-IS) for their support. M. Muehlebach is funded by the German Research Foundation (DFG) under the project 456587626 of the Emmy Noether Programme. C. Vernade is funded by the German Research Foundation (DFG) under both the project 468806714 of the Emmy Noether Programme and under Germany's Excellence Strategy – EXC number 2064/1 – Project number 390727645.

## Impact statement

This paper presents work whose goal is to advance the field of machine learning. There are many potential societal consequences of our work, none which we feel must be specifically highlighted here.

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

## A. Further results on injectivity

The injectivity result of Theorem 4.1 holds only if $\lambda$ is rational and histories are finite. We now justify these constraints with two counterexamples.

**Example A.1** (Irrational $\lambda$). Consider the observation space consisting of $y^1 = (1,0)$ and $y^2 = (0,1)$. The history $h = (y^1, y^2, y^2)$ leads to the memory trace

$$z_\lambda(h) = y^1 + \lambda y^2 + \lambda^2 y^2 = (1, \lambda + \lambda^2).$$

If $\lambda = \phi - 1$, where $\phi = \frac{\sqrt{5}+1}{2}$ is the golden ratio, then $z_\lambda(h') = (1,1)$. By symmetry, the same memory trace will thus be obtained for the history $h' = (y^2, y^1, y^1)$.

**Example A.2** (Infinite history). Consider the same observation space as in Example A.1. Let $h = (y^1, y^2, y^2, \dots)$ and $h' = (y^2, y^1, y^1, \dots)$ be two infinite histories. Then, with $\lambda = \frac{1}{2}$,

$$z_\lambda(h) = \left(1, \sum_{k=1}^{\infty} \frac{1}{2^k}\right) = (1,1).$$

Thus, the traces are equal by symmetry.

Note that the above examples only work because $\lambda \geq \frac{1}{2}$. For $\lambda < \frac{1}{2}$, injectivity is guaranteed by Lemma 4.3. We now extend this result to linearly dependent observation spaces.

**Theorem A.3.** *For all finite sets $\mathcal{Y} \subset \mathbb{Q}^D$, there exists a set $\Lambda \subset \mathbb{Q}$ that is dense in $(0,1)$ with the property that if $\lambda \in \Lambda$, then $z_\lambda$ is injective for finite histories.*

*Proof.* We follow a similar argument as in the proof of Theorem 4.1. Let $h$ and $h'$ be two distinct finite histories that are 0-padded. We write $z \doteq z_\lambda(h)$ and $z' \doteq z_\lambda(h')$. Let $e_1, \dots, e_D$ denote the standard basis of $\mathbb{R}^D$. As $\mathcal{Y}_0 \subset \mathbb{Q}^D$, each $y^i \in \mathcal{Y}_0$ can be written as

$$y^i = \sum_{j=1}^{D} \frac{p_{ij}}{q_{ij}} e_j,$$

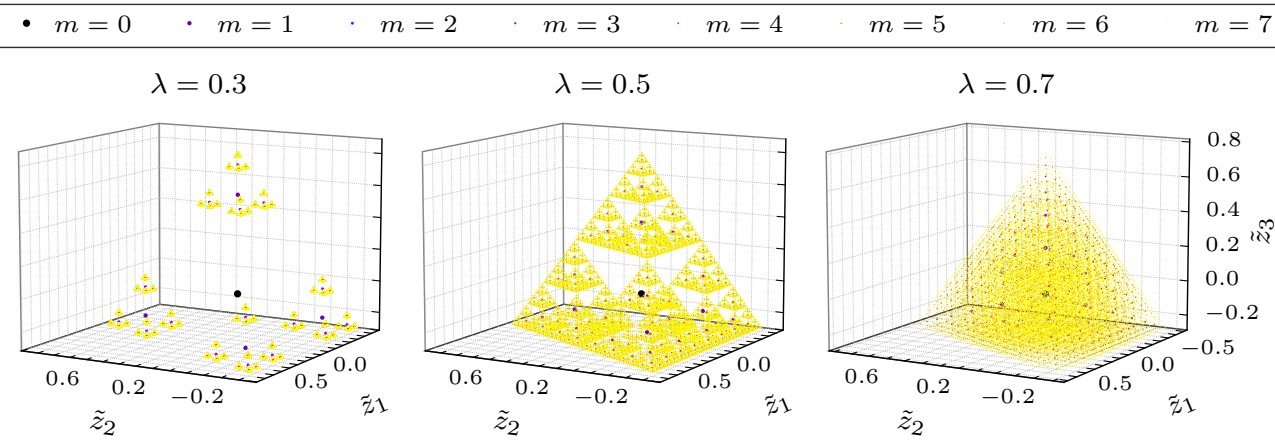

*Figure 6.* A visualization of trace space, where $\mathcal{Y}$ is one-hot with $|\mathcal{Y}| = 4$. To visualize the four-dimensional space, we perform an orthogonal projection onto the three-dimensional space described by (4).

with $p_{ij}$ and $q_{ij}$ coprime integers.

$$\frac{z - z'}{1 - \lambda} = \sum_{i=1}^{|\mathcal{Y}_0|} \alpha_i(\lambda) y^i = \sum_{j=1}^{D} \left\{ \underbrace{\sum_{i=1}^{|\mathcal{Y}_0|} \frac{p_{ij}}{q_{ij}} \alpha_i(\lambda)}_{\beta_j(\lambda)} \right\} e_j,$$

where $\alpha_i$ is defined in (3). By linear independence of the standard basis, we see that $z = z'$ only if $\beta_j(\lambda) = 0$ for all $j \in [D]$. By (3), we can expand $\beta_j$ as

$$\beta_j(\lambda) = \sum_{k=0}^{\infty} \lambda^k \sum_{i=1}^{|\mathcal{Y}_0|} \frac{p_{ij}}{q_{ij}} \big( [y_{-k} = y^i] - [y'_{-k} = y^i] \big).$$

Let $i_k$ and $i'_k$ be the indices (with respect to $\mathcal{Y}_0$) of the observations $k$ steps ago, for $h$ and $h'$, respectively, such that $y_{-k} = y^{i_k}$ and $y'_{-k} = y^{i'_k}$. We can then write $\beta_j$ as

$$\beta_j(\lambda) = \sum_{k=0}^{\infty} \lambda^k \left( \frac{p_{i_k j}}{q_{i_k j}} - \frac{p_{i'_k j}}{q_{i'_k j}} \right).$$

As the histories $h$ and $h'$ are distinct, there must be a $k_0$ such that the corresponding observations are distinct, $y_{-k_0} \neq y'_{-k_0}$. The coefficient of the $\lambda^{k_0}$ term in $\beta_j$ must be nonzero for at least one $j \in [D]$, as we would otherwise have

$$y_{-k_0} - y'_{-k_0} = \sum_{j=1}^{D} \left( \frac{p_{i_{k_0} j}}{q_{i_{k_0} j}} - \frac{p_{i'_{k_0} j}}{q_{i'_{k_0} j}} \right) e_j = 0,$$

which is a contradiction. Thus, we have shown that there exists a $j_0 \in [D]$ such that $\beta_{j_0}$ has at least one nonzero coefficient. In the following we introduce the set $\Lambda$, which we show to be dense in the real numbers, and finally we complete the proof by showing that $\lambda \in \Lambda$ implies $\beta_{j_0}(\lambda) \neq 0$.

The set $\Lambda \subset \mathbb{Q}$ is defined as

$$\Lambda \doteq \left\{ \frac{p}{q} \in \mathbb{Q} \mid \forall i, i' \in [|\mathcal{Y}_0|], j \in [D] : \right.$$

$$\left. \frac{p_{ij}}{q_{ij}} \neq \frac{p_{i'j}}{q_{i'j}} \Rightarrow q \nmid \left( \frac{p_{ij}}{q_{ij}} - \frac{p_{i'j}}{q_{i'j}} \right) \prod_{k=1}^{|\mathcal{Y}_0|} \prod_{l=1}^{D} q_{kl} \right\},$$

where $p$ and $q$ are coprime, and $m \nmid n$ means that $m$ does not divide $n$. We first show that $\Lambda$ is dense in $\mathbb{R}$. Let $a < b$ be two real numbers. As the rational numbers are dense in $\mathbb{R}$, there must exist a number $\frac{\tilde{p}}{\tilde{q}} \in \mathbb{Q}$, with $\tilde{p}$ and $\tilde{q}$ coprime and $\tilde{q} > 1$, such that $a < \frac{\tilde{p}}{\tilde{q}} < b$. (If $\tilde{q} = 1$, we can create a new rational number in $(\tilde{p}, b) \subset (a, b)$ instead.) Let $n > 1$ be a natural number such that

$$\tilde{q}^n > \max \left\{ \hat{c}, \frac{1}{b - \tilde{p}/\tilde{q}} \right\},$$

where

$$\hat{c} \doteq \max_{\substack{i, i' \in [|\mathcal{Y}_0|] \\ j \in [D]}} \left\{ \left( \frac{p_{ij}}{q_{ij}} - \frac{p_{i'j}}{q_{i'j}} \right) \prod_{k=1}^{|\mathcal{Y}_0|} \prod_{l=1}^{D} q_{kl} \right\}.$$

We now define $p \doteq \tilde{p}\tilde{q}^{n-1} + 1$ and $q \doteq \tilde{q}^n$. Then,

$$\frac{p}{q} = \frac{\tilde{p}}{\tilde{q}} + \frac{1}{\tilde{q}^n} \in (a, b).$$

If $p$ and $q$ are coprime, then $\frac{p}{q} \in \Lambda$, as $q$ is too large to divide any of the integers that the definition of $\Lambda$ prohibits it to divide. Let $e$ be a prime factor of $q = \tilde{q}^n$. Then, $e$ is a prime factor of $\tilde{q}$, and hence also of $p - 1 = \tilde{p}\tilde{q}^{n-1}$. But then, $e$ cannot be a prime factor of $p$. Thus, $p$ and $q$ are coprime, and so $\frac{p}{q} \in \Lambda \cap (a, b)$. As $a$ and $b$ were arbitrary real numbers, we have shown that $\Lambda$ is dense in $\mathbb{R}$.

Finally, we show that no $\lambda \in \Lambda$ is a root of $\beta_{j_0}$. Assume, for the sake of contradiction, that there exists a $\lambda = \frac{p}{q} \in \Lambda$,

with $p$ and $q$ coprime, such that $\beta_{j_0}(\lambda) = 0$:

$$\sum_{k=0}^{\infty} \lambda^k \left( \frac{p_{i_k j_0}}{q_{i_k j_0}} - \frac{p_{i'_k j_0}}{q_{i'_k j_0}} \right) = 0$$

$$\iff \sum_{k=0}^{\infty} \lambda^k \underbrace{\left( \frac{p_{i_k j_0}}{q_{i_k j_0}} - \frac{p_{i'_k j_0}}{q_{i'_k j_0}} \right) \prod_{i=1}^{|\mathcal{Y}_0|} \prod_{j=1}^{D} q_{ij}}_{c_k \in \mathbb{Z}} = 0.$$

Then, by the rational root theorem, $q$ divides the highest-order coefficient $c_{\hat{k}}$, which we now know is nonzero. However, the definition of $\Lambda$ prohibits $q$ from dividing an integer of this form. Thus, we know that no $\lambda \in \Lambda$ is a root of $\beta_{j_0}(\lambda)$, and we can conclude that $z \neq z'$. $\qquad\square$

How is the set $\Lambda$ to be interpreted, and why does $\Lambda = \mathbb{Q} \setminus \mathbb{Z}$ does not always work? Consider the observation space $\mathcal{Y} = \{0, 1, 2\} \subset \mathbb{Q}$, which is not linearly independent. If $\lambda = \frac{1}{2}$, then the histories $h = (1, 0)$ and $h' = (0, 2)$ will both produce the memory trace $z = 1$. To build the set $\Lambda$, we first rewrite each observation as $y^i = \frac{p_i}{q_i}$. As all observations are integers, $q_i = 1$ for all three $y^i \in \mathcal{Y}$. Thus, looking at the definition of $\Lambda$, we have for a $\frac{p}{q} \in \mathbb{Q}$, with $p$ and $q$ coprime, that $\frac{p}{q} \in \Lambda$ if and only if,

$$\forall i, i' \in [|\mathcal{Y}_0|] : \left\{ \frac{p_i}{q_i} \neq \frac{p_{i'}}{q_{i'}} \Rightarrow q \setminus \left( \frac{p_i}{q_i} - \frac{p_{i'}}{q_{i'}} \right) \prod_{k=1}^{|\mathcal{Y}_0|} q_k \right\}$$

$$\iff \forall i, i' \in [|\mathcal{Y}_0|] : \left\{ p_i \neq p_{i'} \Rightarrow q \setminus (p_i - p_{i'}) \right\}$$

$$\iff \forall d \in \{\pm 1, \pm 2\} : q \setminus d$$

$$\iff q \setminus 2$$

We thus see that $\Lambda = \mathbb{Q} \setminus \frac{1}{2}\mathbb{Z}$. Thus, for $0 < \lambda < 1$, our choice of $\lambda = \frac{1}{2}$ was the only point where $\Lambda$ differs from $\mathbb{Q} \setminus \mathbb{Z}$.

A relevant special case of this result is concerned with 0-1-encodings of the observations. The next result shows that, in this case, the set $\Lambda$ is identical to the one in Theorem 4.1.

**Corollary A.4.** *If $\mathcal{Y} \subset \{0, 1\}^D$ and $\lambda \in \mathbb{Q} \setminus \mathbb{Z}$, then $z_\lambda$ is injective for finite histories.*

*Proof.* We will show that the set $\Lambda$ that is constructed in the proof of Theorem A.3 is exactly $\Lambda = \mathbb{Q} \setminus \mathbb{Z}$ for this choice of $\mathcal{Y}$. All $y^i \in \mathcal{Y}_0$ can be written as

$$y^i = \sum_{j=1}^{D} \frac{p_{ij}}{q_{ij}} e_j,$$

with $q_{i,j} = 1$ and $p_{i,j} \in \{0, 1\}$ for all $i \in [|\mathcal{Y}_0|]$ and $j \in D$. Thus, if $\lambda = \frac{p}{q} \in \mathbb{Q}$, with $p$ and $q$ coprime, then $\lambda \in \Lambda$ if and only if,

$$\forall i, i' \in [|\mathcal{Y}_0|], j \in [D] :$$
$$\left\{ \frac{p_{ij}}{q_{ij}} \neq \frac{p_{i'j}}{q_{i'j}} \Rightarrow q \setminus \left( \frac{p_{ij}}{q_{ij}} - \frac{p_{i'j}}{q_{i'j}} \right) \prod_{k=1}^{|\mathcal{Y}_0|} \prod_{l=1}^{D} q_{k,l} \right\}$$

$$\iff \forall i, i' \in [|\mathcal{Y}_0|], j \in [D] :$$
$$\{p_{ij} \neq p_{i'j} \Rightarrow q \setminus (p_{ij} - p_{i'j})\}$$

$$\iff \forall d \in \{-1, 1\} : q \setminus d$$

$$\iff q \setminus 1.$$

Thus $\Lambda = \mathbb{Q} \setminus \mathbb{Z}$. $\qquad\square$

## B. Proofs

In this section we present all proofs that are omitted in the main text.

**Lemma 4.2** (Concentration). *Let $h$ and $\bar{h}$ be two histories of one-hot observations such that $\mathrm{win}_m(h) = \mathrm{win}_m(\bar{h})$ for some $m$. Then, the corresponding traces satisfy*

$$\|z_\lambda(h) - z_\lambda(\bar{h})\| \leq \sqrt{2}\lambda^m.$$

*Proof.* Without loss of generality we can assume the histories are infinite (by applying 0-padding). By expanding the distance, we get

$$\|z_\lambda(h) - z_\lambda(\bar{h})\|$$
$$= (1 - \lambda) \left\| \sum_{k=m}^{\infty} \lambda^k (y_{-k} - \bar{y}_{-k}) \right\|$$
$$\leq (1 - \lambda)\lambda^m \sum_{k=0}^{\infty} \lambda^k \|y_{-k-m} - \bar{y}_{-k-m}\|$$
$$\leq \sqrt{2}\lambda^m. \qquad\square$$

**Lemma 4.3** (Separation). *Let $h$ and $\bar{h}$ be two histories of one-hot observations such that $\mathrm{win}_m(h) \neq \mathrm{win}_m(\bar{h})$ for some $m$. Then, if $\lambda \leq \frac{1}{2}$, the corresponding traces satisfy*

$$\|z_\lambda(h) - z_\lambda(\bar{h})\| \geq \sqrt{2}(1 - 2\lambda)\lambda^{m-1}.$$

*Proof.* We write $z_\lambda^m \doteq z_\lambda \circ \mathrm{win}_m$. Without loss of generality we can assume the histories are infinite (by applying 0-padding). By making use of the reverse triangle inequality, we get

$$\|z_\lambda(h) - z_\lambda(\bar{h})\|$$
$$= (1 - \lambda) \left\| \sum_{k=0}^{\infty} \lambda^k (y_{-k} - \bar{y}_{-k}) \right\|$$
$$= \left\| z_\lambda^m(h) - z_\lambda^m(\bar{h}) \right.$$
$$\left. + (1 - \lambda) \sum_{k=m}^{\infty} \lambda^k (y_{-k} - \bar{y}_{-k}) \right\|$$
$$\geq \left| \|z_\lambda^m(h) - z_\lambda^m(\bar{h})\| \right.$$
$$\left. - (1 - \lambda) \left\| \sum_{k=m}^{\infty} \lambda^k (y_{-k} - \bar{y}_{-k}) \right\| \right|.$$

We will show that the first term is bounded below by

$$\|z_\lambda^m(h) - z_\lambda^m(\bar{h})\| \geq \sqrt{2}(1 - \lambda)\lambda^{m-1},$$

whereas the second term can be upper-bounded using the triangle inequality and the fact that $\|y_{-k} - \bar{y}_{-k}\| \leq \sqrt{2}$:

$$(1-\lambda)\Big\|\sum_{k=m}^{\infty} \lambda^k(y_{-k} - \bar{y}_{-k})\Big\|$$
$$\leq \sqrt{2}\lambda^m \leq \sqrt{2}(1-\lambda)\lambda^{m-1},$$

where we used $\lambda \leq \frac{1}{2}$ in the last step. This allows us to drop the absolute value in our bound above and get

$$\|z_\lambda(h) - z_\lambda(\bar{h})\| \geq \sqrt{2}(1-\lambda)\lambda^{m-1} - \sqrt{2}\lambda^m$$
$$= \sqrt{2}(1-2\lambda)\lambda^{m-1}$$

as desired.

To prove the lower bound on the first term, we perform induction on $m$. For the base case $m = 1$ (a single observation), $\mathrm{win}_m(h) \neq \mathrm{win}_m(\bar{h})$ implies that $y \neq \bar{y}$. We thus have $\|z_\lambda^1(h) - z_\lambda^1(\bar{h})\| = \sqrt{2}(1-\lambda)$, satisfying the desired inequality.

We now suppose that the result holds for some $m \in \mathbb{N}$, and consider the case of two histories $h$ and $\bar{h}$ such that $\mathrm{win}_{m+1}(h) \neq \mathrm{win}_{m+1}(\bar{h})$. Using the reverse triangle inequality, we have

$$\|z_\lambda^{m+1}(h) - z_\lambda^{m+1}(\bar{h})\|$$
$$= (1-\lambda)\Big\|\sum_{k=0}^{m} \lambda^k(y_{-k} - \bar{y}_{-k})\Big\|$$
$$= \|(1-\lambda)(y - \bar{y}) + \lambda\{z_\lambda^m(h_{-1}) - z_\lambda^m(\bar{h}_{-1})\}\|$$
$$\geq \big|(1-\lambda)\|y - \bar{y}\| - \lambda\|z_\lambda^m(h_{-1}) - z_\lambda^m(\bar{h}_{-1})\|\big|, \quad (6)$$

There are two cases to consider. First, suppose that $y = \bar{y}$, such that the first term in (6) is $(1-\lambda)\|y - \bar{y}\| = 0$. In this case, we know that $\mathrm{win}_m(h_{-1}) \neq \mathrm{win}_m(\bar{h}_{-1})$, since otherwise $\mathrm{win}_{m+1}(h) = \mathrm{win}_{m+1}(\bar{h})$. Thus, the induction hypothesis and (6) guarantee that

$$\|z_\lambda^{m+1}(h) - z_\lambda^{m+1}(\bar{h})\| \geq \lambda\|z_\lambda^m(h_{-1}) - z_\lambda^m(\bar{h}_{-1})\|$$
$$\geq \sqrt{2}(1-\lambda)\lambda^m.$$

Now, suppose that $y \neq \bar{y}$, such that $\|y - \bar{y}\| = \sqrt{2}$. We can upper-bound the second term in (6) by making use of the triangle inequality and the fact that $\lambda \leq (1-\lambda)$:

$$\lambda\|z_\lambda^m(h_{-1}) - z_\lambda^m(\bar{h}_{-1})\|$$
$$= \lambda(1-\lambda)\Big\|\sum_{k=0}^{m-1} \lambda^k(y_{-k-1} - \bar{y}_{-k-1})\Big\|$$
$$\leq \sqrt{2}(1-\lambda^m)\lambda \leq \sqrt{2}\lambda \leq \sqrt{2}(1-\lambda).$$

Thus, the first term of (6) dominates the second term, and we can drop the absolute value:

$$\|z_\lambda^{m+1}(h) - z_\lambda^{m+1}(\bar{h})\|$$
$$\geq (1-\lambda)\|y - \bar{y}\| - \lambda\|z_\lambda^m(h_{-1}) - z_\lambda^m(\bar{h}_{-1})\|$$
$$\geq \sqrt{2}(1-\lambda) - \sqrt{2}(1-\lambda^m)\lambda$$
$$\geq \sqrt{2}(1-\lambda) - \sqrt{2}(1-\lambda^m)(1-\lambda)$$
$$= \sqrt{2}(1-\lambda)\lambda^m,$$

where we have again used that $\lambda \leq \frac{1}{2}$. $\qquad\square$

**Lemma 4.4.** *If $\mathcal{Y}$ is one-hot, then the Minkowski dimension of $\mathcal{Z}_\lambda$ is, for all $\lambda < \frac{1}{2}$,*

$$\dim(\mathcal{Z}_\lambda) = \frac{\log|\mathcal{Y}|}{\log(1/\lambda)} \doteq d_\lambda.$$

*For all $\lambda \in [0, 1)$, we have $\dim(\mathcal{Z}_\lambda) \leq \min\{|\mathcal{Y}| - 1, d_\lambda\}$.*

*Proof.* The set $\mathcal{Z}_\lambda$ is a *self-similar fractal*, which means that it is composed of several scaled and translated copies of itself. In particular,

$$\mathcal{Z}_\lambda = \lambda\mathcal{Z}_\lambda + (1-\lambda)\mathcal{Y},$$

where the scalar multiplication is applied to every element of the set, and the addition is the *Minkowski addition* which adds all pairs of points of the two sets. Thus, $\mathcal{Z}_\lambda$ consists of precisely $|\mathcal{Y}|$ copies of itself, each scaled by $\lambda$. If $\lambda < \frac{1}{2}$, then Lemma 4.3 guarantees that these $|\mathcal{Y}|$ smaller copies do not overlap (this is also visualized in Fig. 2). The results now all follow directly from known properties of the Minkowski dimension (Tao, 2010, Section 1.15.1). The upper bound of $|\mathcal{Y}| - 1$ follows from (4). $\qquad\square$

**Theorem 5.1** (Hoeffding bound)**.** *Given a dataset of $n$ trajectories from an environment $\mathcal{E}$, a function class $\mathcal{F}$, and some $\epsilon > 0$, let $\mathcal{F}^\epsilon$ be the smallest $\epsilon$-cover of $\mathcal{F}$ and $f_n \doteq \arg\min_{f \in \mathcal{F}^\epsilon} \mathcal{R}_n(f)$. Then, with probability at least $1 - \delta$,*

$$\mathcal{R}_\mathcal{E}(f_n) \leq \mathcal{R}_\mathcal{E}(\mathcal{F}) + \Delta^2\sqrt{\frac{H_\epsilon(\mathcal{F}) + \log\frac{2}{\delta}}{2n}} + \epsilon\Delta + \frac{\epsilon^2}{2},$$

*where we have defined $\Delta \doteq \bar{v} - \underline{v}$.*

*Proof.* In the language of statistical learning, the return error $\mathcal{R} \doteq \mathcal{R}_\mathcal{E}$ is a measure of *risk*, and the empirical return error $\mathcal{R}_n$ is the corresponding *empirical risk*. The argument presented here is standard (cf. Bousquet et al., 2003, Section 3.5). Let $f_\epsilon \doteq \arg\min_{f \in \mathcal{F}^\epsilon} \mathcal{R}(f)$. We have,

$$\mathcal{R}(f_n) - \mathcal{R}(\mathcal{F}^\epsilon) = \mathcal{R}(f_n) - \mathcal{R}_n(f_n) + \mathcal{R}_n(f_n) - \mathcal{R}(f_\epsilon)$$
$$\leq \mathcal{R}(f_n) - \mathcal{R}_n(f_n) + \mathcal{R}_n(f_\epsilon) - \mathcal{R}(f_\epsilon)$$
$$\leq 2\max_{f \in \mathcal{F}^\epsilon}|\mathcal{R}(f) - \mathcal{R}_n(f)|. \quad (7)$$

Hoeffding's inequality states that if $\{\xi_i\}_{i=1}^n$ are $n$ independent random variables with mean $\mu$ that are contained almost surely in the interval $[a, b]$, then

$$\mathbb{P}\left\{\left|\frac{1}{n}\sum_{i=1}^n \xi_i - \mu\right| \geq \eta\right\} \leq 2\exp\left(-\frac{2n\eta^2}{(b-a)^2}\right)$$

for any $\eta > 0$. We can apply this inequality to our setting by defining $\xi_i(f) \doteq \frac{1}{2}\left\{f(y_0, y_{-1}, \ldots) - \sum_{t=0}^\infty \gamma^t r(y_{t+1})\right\}^2$ for $f \in \mathcal{F}^\epsilon$. Then, $\xi_i(f) \in [0, \Delta^2/2]$ almost surely for any $f$. These variables are independent since the dataset is i.i.d., so Hoeffding's inequality applies. In particular, we see that

$$\mathcal{R}_n(f) = \frac{1}{n}\sum_{i=1}^n \xi_i(f) \quad \text{and} \quad \mathcal{R}(f) = \mathbb{E}\big[\mathcal{R}_n(f)\big] = \mu(f).$$

Thus, for $\eta > 0$, we have

$$\mathbb{P}\left\{2\max_{f \in \mathcal{F}^\epsilon}|\mathcal{R}(f) - \mathcal{R}_n(f)| \geq \eta\right\}$$
$$= \mathbb{P}\bigcup_{f \in \mathcal{F}^\epsilon}\{|\mathcal{R}(f) - \mathcal{R}_n(f)| \geq \eta/2\}$$
$$\leq \sum_{f \in \mathcal{F}^\epsilon}\mathbb{P}\{|\mathcal{R}(f) - \mathcal{R}_n(f)| \geq \eta/2\}$$
$$\leq 2|\mathcal{F}^\epsilon|\exp\left(-\frac{2n\eta^2}{\Delta^4}\right) \doteq \delta.$$

Solving for $\eta$, we get that with probability at least $1 - \delta$,

$$2\max_{f \in \mathcal{F}^\epsilon}|\mathcal{R}(f) - \mathcal{R}_n(f)| \leq \eta \doteq \Delta^2\sqrt{\frac{\log|\mathcal{F}^\epsilon| + \log\frac{2}{\delta}}{2n}}.$$

The final result follows from (7), Lemma 5.2 (where we let $\mathcal{G} \doteq \mathcal{F}^\epsilon$), and the definition of metric entropy. $\quad\square$

**Lemma 5.2.** *Let $\mathcal{E}$ be an environment, $\mathcal{F}$ a function class, and $\epsilon > 0$. If $\mathcal{G}$ is an $\epsilon$-cover of $\mathcal{F}$, then*

$$\mathcal{R}_\mathcal{E}(\mathcal{G}) \leq \mathcal{R}_\mathcal{E}(\mathcal{F}) + \epsilon\Delta + \frac{\epsilon^2}{2}.$$

*Proof.* Define $f \in \mathcal{F}$ as $f \doteq \arg\min_{f' \in \mathcal{F}}\mathcal{R}_\mathcal{E}(f')$ and let $g \in \mathcal{G}$ be such that $\|f - g\|_\infty \leq \epsilon$. Then,

$$2\mathcal{R}_\mathcal{E}(\mathcal{G}) \leq 2\mathcal{R}_\mathcal{E}(g)$$
$$= \mathbb{E}_\mathcal{E}\left[\left\{g(h_0) - \overbrace{\sum_{t=0}^\infty \gamma^t r(y_{t+1})}^{R_0}\right\}^2\right]$$
$$= \mathbb{E}_\mathcal{E}\left[\{g(h_0) - f(h_0) + f(h_0) - R_0\}^2\right]$$
$$\leq \mathbb{E}_\mathcal{E}\left[\{\epsilon + |f(h_0) - R_0|\}^2\right]$$
$$= \mathbb{E}_\mathcal{E}\left[\epsilon^2 + 2\epsilon|f(h_0) - R_0| + |f(h_0) - R_0|^2\right]$$
$$\leq 2\mathcal{R}_\mathcal{E}(\mathcal{F}) + 2\epsilon\Delta + \epsilon^2. \quad\square$$

**Lemma 5.3.** *Let $m \in \mathbb{N}_0$ be a window length. Then, the metric entropy of $\mathcal{F}_m$ is, for all $\epsilon > 0$,*

$$H_\epsilon(\mathcal{F}_m) = |\mathcal{Y}|^m \log\left\lceil\frac{\Delta}{2\epsilon}\right\rceil.$$

*Thus, as a function of $m$, $H_\epsilon(\mathcal{F}_m) \in \Theta(|\mathcal{Y}|^m)$.*

*Proof.* An $\epsilon$-*packing* of a set $\mathcal{F}$ is a finite subset $\mathcal{G}^\epsilon \subset \mathcal{F}$ with the property that, for any $f, g \in \mathcal{G}^\epsilon$, it holds that $\|f - g\|_\infty \geq \epsilon$. The $\epsilon$-*packing number* $M_\epsilon(\mathcal{F})$ of $\mathcal{F}$ is the cardinality of the largest $\epsilon$-packing of $\mathcal{F}$. This number is closely related to the covering number $N_\epsilon(\mathcal{F})$ defined earlier. Let $\epsilon > 0$ and let $\mathcal{F}^\epsilon$ and $\mathcal{G}^{2\epsilon}$ be an $\epsilon$-cover and a $2\epsilon$-packing of the set $\mathcal{F}$, respectively. Then, it holds that (Tao, 2014)

$$|\mathcal{G}^{2\epsilon}| \leq M_{2\epsilon}(\mathcal{F}) \leq N_\epsilon(\mathcal{F}) \leq |\mathcal{F}^\epsilon|. \qquad (8)$$

We will prove the result by finding a set $\mathcal{F}_m^\epsilon$, for each $m \in \mathbb{N}_0$ and each $\epsilon > 0$, that is simultaneously an $\epsilon$-cover and a $2\epsilon$-packing of $\mathcal{F}_m$. By the inequality above, this implies that $H_\epsilon(\mathcal{F}_m) = \log|\mathcal{F}_m^\epsilon|$. The set we consider is

$$\mathcal{F}_m^\epsilon \doteq \{f \circ \text{win}_m \mid f : \mathcal{Y}^m \to \mathcal{V}_\epsilon\} \subset \mathcal{F}_m,$$

where $\mathcal{V}_\epsilon$ is a set of $\lceil\Delta/(2\epsilon)\rceil$ points in $[\underline{v}, \bar{v}]$ that both $\epsilon$-covers this interval and $2\epsilon$-packs it. The set $\mathcal{V}_\epsilon$ exists because we can fit $\lceil\Delta/(2\epsilon)\rceil$ uniformly spaced points with equal distance $2\epsilon$ into the interval, since this only requires a length of $\lceil\Delta/(2\epsilon) - 1\rceil(2\epsilon) < \Delta$. Taking these points as the centers of $\epsilon$-balls, we see that the total volume covered by these balls is $\lceil\Delta/(2\epsilon)\rceil(2\epsilon) \geq \Delta$, since there is no overlap. Thus, if placed at an appropriate position in $[\underline{v}, \bar{v}]$, these points both cover an pack the interval as desired.

Now consider the set $\mathcal{F}_m^\epsilon$. Let $f, g \in \mathcal{F}_m^\epsilon$ be two distinct functions. Then, there must be some history $h$ such that $f(h) \neq g(h)$. As $f(h), g(h) \in \mathcal{V}_\epsilon$, this implies that $|f(h) - g(h)| \geq 2\epsilon$. In other words, $\|f - g\|_\infty \geq 2\epsilon$, and so $\mathcal{F}_m^\epsilon$ is a $2\epsilon$-packing of $\mathcal{F}_m$. Now let $f \in \mathcal{F}_m$. Then, for every history $h$, there exists a value $\tilde{v} \in \mathcal{V}_\epsilon$ such that $|\tilde{v} - f(h)| \leq \epsilon$ (since $\mathcal{V}_\epsilon$ covers $[\underline{v}, \bar{v}]$). Define $g \in \mathcal{F}_m^\epsilon$ such that $g(h)$ is exactly this $\tilde{v} \in \mathcal{V}_\epsilon$ for each history $h$. Then, $\|f - g\|_\infty \leq \epsilon$, and so $\mathcal{F}_m^\epsilon$ is an $\epsilon$-cover of $\mathcal{F}_m$. Thus, we have

$$H_\epsilon(\mathcal{F}_m) = \log|\mathcal{F}_m^\epsilon| = \log|\mathcal{V}_\epsilon|^{|\mathcal{Y}^m|} = |\mathcal{Y}|^m \log\lceil\Delta/(2\epsilon)\rceil.$$
$\square$

**Lemma 5.7.** *Let $|\mathcal{Y}| > 1$. If $\lambda < \frac{1}{2}$, then $d_\lambda < |\mathcal{Y}| - 1$.*

*Proof.* As $|\mathcal{Y}| \geq 2$, we have

$$\left(1 + \frac{1}{|\mathcal{Y}|}\right)^{|\mathcal{Y}|} < e < |\mathcal{Y}| + 1$$
$$\implies \left(\frac{|\mathcal{Y}|}{|\mathcal{Y}| + 1}\right)^{|\mathcal{Y}|} > (|\mathcal{Y}| + 1)^{-1}$$

$$\Longleftrightarrow \qquad |\mathcal{Y}| > (|\mathcal{Y}|+1)^{\frac{|\mathcal{Y}|-1}{|\mathcal{Y}|}}$$

$$\Longleftrightarrow \qquad |\mathcal{Y}|^{\frac{-1}{|\mathcal{Y}|-1}} < (|\mathcal{Y}|+1)^{\frac{-1}{|\mathcal{Y}|}}.$$

Thus, $|\mathcal{Y}|^{\frac{-1}{|\mathcal{Y}|-1}}$ is increasing in $|\mathcal{Y}|$, and therefore, as $|\mathcal{Y}| \geq 2$, the minimum is achieved when $|\mathcal{Y}| = 2$, which yields $2^{\frac{-1}{2-1}} = \frac{1}{2}$. From this, it follows that

$$\lambda < \frac{1}{2}$$

$$\Longrightarrow \qquad \lambda < |\mathcal{Y}|^{\frac{-1}{|\mathcal{Y}|-1}}$$

$$\Longleftrightarrow \quad \log(1/\lambda) > \frac{\log|\mathcal{Y}|}{|\mathcal{Y}|-1}$$

$$\Longleftrightarrow \quad \frac{\log|\mathcal{Y}|}{\log(1/\lambda)} < |\mathcal{Y}|-1$$

$$\Longleftrightarrow \qquad d_\lambda > |\mathcal{Y}|-1. \qquad \square$$

**Theorem 5.8** (T-maze). *There exists a sequence $(\mathcal{E}_k)$ of environments (with constant observation space $\mathcal{Y}$) with the property that, for every $\epsilon > 0$ and every $k \in \mathbb{N}$,*

$$\min_m \{H_\epsilon(\mathcal{F}_m) \mid \mathcal{R}_{\mathcal{E}_k}(\mathcal{F}_m) = 0\} \in \Omega(|\mathcal{Y}|^k), \text{ and}$$

$$\min_{\lambda,L} \{H_\epsilon(\mathcal{F}_{\lambda,L}) \mid \mathcal{R}_{\mathcal{E}_k}(\mathcal{F}_{\lambda,L}) = 0\} \in \mathcal{O}(k^{|\mathcal{Y}|-1}).$$

*Proof.* The *T-maze* environment $\mathcal{E}_k$ with corridor length $k$ is defined as follows (see Fig. 1). There are $|\mathcal{Y}| = 5$ possible observations, $\mathcal{Y} = \{\mathtt{a}, \mathtt{b}, \mathtt{o}, \mathtt{x}, \mathtt{y}\}$, each encoded as a one-hot vector in $\mathbb{R}^5$. An episode starts at the left of the corridor, where the agent receives either observation $y_0 = \mathtt{a}$ or $y_0 = \mathtt{b}$, each with equal probability. Moving along the corridor to the right, the agent receives the observation $\mathtt{o}$ at every step until, at time $t = k - 1$, it reaches the end of the corridor, where the observation $y_{k-1}$ is, with equal probability, either $\mathtt{x}$ or $\mathtt{y}$. At this point, the agent has to decide whether to go up ($\uparrow$) or down ($\downarrow$), after which the episode ends. The reward it receives in the final step is determined by the following table.

Table 1. Rewards in the T-maze

|  |  | $y_0 = \mathtt{a}$ | $y_0 = \mathtt{b}$ |
|---|---|---|---|
| $y_{k-1} = \mathtt{x}$ | $u_{k-1} = \uparrow$ | $+1$ | $-1$ |
|  | $u_{k-1} = \downarrow$ | $-1$ | $+1$ |
| $y_{k-1} = \mathtt{y}$ | $u_{k-1} = \uparrow$ | $-1$ | $+1$ |
|  | $u_{k-1} = \downarrow$ | $+1$ | $-1$ |

All other transitions give a reward of 0. We consider the fixed policy that always goes up ($\uparrow$) at the end of the corridor. Our goal is to estimate the value function of this policy. The true value function is given by the following table (where histories are written left-to-right and $i \in \{0, \ldots, k-2\}$).

Table 2. T-maze value function for 'always-up' policy

| History $h$ | $v(h)$ |
|---|---|
| $\mathtt{ao}^i$ | $0$ |
| $\mathtt{bo}^i$ | $0$ |
| $\mathtt{ao}^{k-2}\mathtt{x}$ | $+1$ |
| $\mathtt{bo}^{k-2}\mathtt{x}$ | $-1$ |
| $\mathtt{ao}^{k-2}\mathtt{y}$ | $-1$ |
| $\mathtt{bo}^{k-2}\mathtt{y}$ | $+1$ |
| other | $\perp$ |

Before observing $y_{k-1}$, the value must be 0, since both $\mathtt{x}$ and $\mathtt{y}$ are equally likely, and $u_{k-1} = \uparrow$ will thus result in a reward of $+1$ or $-1$ with equal probability. All histories other than the ones in the table above can have an arbitrary value (denoted $v(h) = \perp$ above), since the probability of these histories is 0. To represent this value function accurately with a length-$m$ window (where we consider the 0-padded histories), it is necessary that $m \geq k$, since it would otherwise be impossible to differentiate between $\mathtt{ao}^{k-2}\mathtt{x}$ and $\mathtt{bo}^{k-2}\mathtt{x}$. This already gives us the first result. Using Lemma 5.3 (with $\Delta = 2$), $m \geq k$ implies that, for all $\epsilon > 0$,

$$H_\epsilon(\mathcal{F}_m) \geq |\mathcal{Y}|^k \log\lceil 1/\epsilon \rceil \in \Omega(|\mathcal{Y}|^k).$$

We now construct a function in $\mathcal{F}_{\lambda,L}$, where $\lambda = \frac{k-1}{k}$ and $L = \sqrt{2}ek$ that achieves a return error of 0. This implies, by Lemma 5.4, that for all $\epsilon > 0$,

$$H_\epsilon(\mathcal{F}_{\lambda,L}) \leq \log\left\lceil \frac{2}{\epsilon} \right\rceil \left\lceil \frac{2\sqrt{2}ek\sqrt{|\mathcal{Y}|-1}}{\epsilon} \right\rceil^{|\mathcal{Y}|-1} \in \mathcal{O}(k^{|\mathcal{Y}|-1}).$$

We define $f \doteq \bar{f} \circ z_\lambda$ by setting $\hat{f}(z_\lambda(h)) \doteq v(h)$ for each (0-padded) history $h$ in Table 2 and extending $\hat{f}$ to a $\mathrm{Lip}(\hat{f})$-Lipschitz continuous function $\bar{f} : [0,1]^{|\mathcal{Y}|} \to [-1,1]$ using Kirszbraun's theorem. This ensures that $f$ accurately represents the value function. To complete the proof, we need to verify that $\mathrm{Lip}(\hat{f}) \leq L$. We thus check all pairs of histories in Table 2 whose values are not equal. Checking the first pair yields

$$\|z_\lambda(\mathtt{ao}^{k-2}\mathtt{x}) - z_\lambda(\mathtt{bo}^{k-2}\mathtt{x})\|$$
$$= \sqrt{2}(1-\lambda)\lambda^{k-1} = \frac{\sqrt{2}}{k\left(1+\frac{1}{k-1}\right)^{k-1}} \geq \frac{\sqrt{2}}{ek}.$$

This is equal to $\|z_\lambda(\mathtt{ao}^{k-2}\mathtt{y}) - z_\lambda(\mathtt{bo}^{k-2}\mathtt{y})\|$ by symmetry. Checking the next pair yields

$$\|z_\lambda(\mathtt{ao}^{k-2}\mathtt{x}) - z_\lambda(\mathtt{ao}^{k-2}\mathtt{y})\|$$
$$= (1-\lambda)\|\mathtt{x} - \mathtt{y}\| = \sqrt{2}(1-\lambda) = \sqrt{2}/k,$$

which is equal to $\|z_\lambda(\mathtt{bo}^{k-2}\mathtt{x}) - z_\lambda(\mathtt{bo}^{k-2}\mathtt{y})\|$ by symmetry. Checking the next pair yields

$$\|z_\lambda(\mathtt{ao}^{k-2}\mathtt{x}) - z_\lambda(\mathtt{ao}^i)\|$$
$$= \|(1-\lambda)\mathtt{x} + \underbrace{\lambda z_\lambda(\mathtt{ao}^{k-2}) - z_\lambda(\mathtt{ao}^i)}_{\perp (1-\lambda)\mathtt{x}}\|$$
$$\geq (1-\lambda) = 1/k,$$

where we have used the Pythagorean theorem. This is equal to $\|z_\lambda(\mathtt{ao}^{k-2}\mathtt{y}) - z_\lambda(\mathtt{ao}^i)\|$, $\|z_\lambda(\mathtt{bo}^{k-2}\mathtt{x}) - z_\lambda(\mathtt{bo}^i)\|$, and $\|z_\lambda(\mathtt{bo}^{k-2}\mathtt{y}) - z_\lambda(\mathtt{bo}^i)\|$ by symmetry. Similarly,

$$\|z_\lambda(\mathtt{ao}^{k-2}\mathtt{x}) - z_\lambda(\mathtt{bo}^i)\|$$
$$= \|(1-\lambda)\mathtt{x} + \underbrace{\lambda z_\lambda(\mathtt{ao}^{k-2}) - z_\lambda(\mathtt{bo}^i)}_{\perp (1-\lambda)\mathtt{x}}\|$$
$$\geq (1-\lambda) = 1/k.$$

This is equal to $\|z_\lambda(\mathtt{ao}^{k-2}\mathtt{y}) - z_\lambda(\mathtt{bo}^i)\|$, $\|z_\lambda(\mathtt{bo}^{k-2}\mathtt{x}) - z_\lambda(\mathtt{ao}^i)\|$, and $\|z_\lambda(\mathtt{bo}^{k-2}\mathtt{y}) - z_\lambda(\mathtt{ao}^i)\|$ by symmetry. Finally,

$$\|z_\lambda(\mathtt{ao}^{k-2}\mathtt{x}) - z_\lambda(\mathtt{bo}^{k-2}\mathtt{y})\|$$
$$= \|(1-\lambda)(\mathtt{x} - \mathtt{y}) + \underbrace{\lambda\{z_\lambda(\mathtt{ao}^{k-2}) - z_\lambda(\mathtt{bo}^{k-2})\}}_{\perp (1-\lambda)(\mathtt{x}-\mathtt{y})}\|$$
$$\geq \sqrt{2}(1-\lambda) = \sqrt{2}/k$$

which is equal to $\|z_\lambda(\mathtt{ao}^{k-2}\mathtt{y}) - z_\lambda(\mathtt{bo}^{k-2}\mathtt{x})\|$ by symmetry. The Lipschitz constant of $\hat{f}$ now satisfies $\mathrm{Lip}(\hat{f}) \leq 2/d$, where $d = \sqrt{2}/(ek)$ is the smallest of the distances above. Thus, we have $\mathrm{Lip}(\hat{f}) \leq L$ as desired. $\qquad\square$

## C. Implementation details

Our implementations of both TD learning and PPO, as well as the two environments (Sutton's noisy random walk and the T-maze) are available online at https://github.com/onnoeberhard/memory-traces.

Our PPO implementation is based on Huang et al. (2022), and the Minigrid T-maze environment implementation uses the xminigrid library (Nikulin et al., 2024). The hyperparameters we used in our PPO experiments are compiled in Table 3. We performed a hyperparameter search over the discount factor $\gamma \in \{0.9, 0.99, 0.999\}$ and plot the results using the best performing values in Fig. 5. Both actor and critic have a neural network architecture of two hidden layers with 64 neurons each and tanh activation functions. We used the Adam optimizer with a linear learning rate decay from 0.0003 to 0 as suggested by Huang et al. (2022).

Our TD experiments ran for $100,000$ steps and we performed a hyperparameter search for the step size $\alpha$ over a range of 13 logarithmically spaced values between 0.0001 and 1.0. In Fig. 4, we show the results using the best step size for each individual value of the window length $m$, but

keep the step size constant at $\alpha = 0.02$ for all values of $\lambda$ for the memory trace. This is the reason why it looks like memory traces are slightly worse with $\lambda = 0$ than windows with $m = 1$; this discrepancy would disappear if we also optimized the step sizes in the memory trace experiments.

*Table 3.* PPO hyperparameters

| Parameter | Value |
|---|---|
| Total number of steps | $1,024,000,000$ |
| Number of parallel environments | 16 |
| Number of steps per update | $128 \times 16$ |
| Learning rate | $0.0003 \to 0$ |
| Generalized advantage estimation $\lambda$ | 0.95 |
| Number of epochs | 2 |
| Number of minibatches | 8 |
| Clipping parameter $\epsilon$ | 0.2 |
| Value loss weight | 0.5 |
| Entropy coefficient | 0.01 |
| Maximum gradient norm | 0.5 |

