# OpenReview forum: "Partially Observable Reinforcement Learning with Memory Traces"
_ICML.cc/2025/Conference — ICML 2025 poster_

### Official Review · Reviewer_hzft · 2025-03-09

**Overall Recommendation:** 4

**Summary:**

This paper addresses the problem of history summarization in POMDPs. Specifically, it proposes a new method to summarize the history with memory traces, which inspired by eligibility traces, in place of the ubiquitous finite-length window. Memory traces induce a representation that accounts for the whole history of observations with exponential discounting of the past. The paper first studies conditions for which the memory trace is an injective function of the history through an original geometric perspective. Then, it compares the properties of memory traces, namely estimation error and complexity, against finite windows in the problem of offline on-policy evaluation, showing that memory traces are competitive with windows when the forgetting is fast and potentially better than windows when the forgetting is slow. Finally, it provides an empirical evaluation in both a toy domain and an illustrative deep RL benchmark to back up the theory.

**Claims And Evidence:**

The claims are supported by theoretical derivations and a brief empirical analysis.

**Essential References Not Discussed:**

The paper reports only a few references of a very large body of work on POMDPs. While a broader account of previous POMDP literature would be nice, I am not aware of any critical reference that should be discussed. The method is remarkably simple and I would not be surprised if someone came up with a similar ideas in prior works. There is a lot of works on eligibility traces in POMDP, but hard to tell whether they are related at all. I am more confident on the novelty of the reported analysis, as it employs arguments and techniques that I did not see in prior works.

**Experimental Designs Or Analyses:**

The experiments are conducted in simple domains, for which the considered design makes sense.

**Methods And Evaluation Criteria:**

Comparing the properties of different history summarization techniques in terms of policy evaluation makes sense. The choice empirical domains also fits the theoretical/conceptual nature of the work.

**Other Comments Or Suggestions:**

The work is really nice and original! I am happy to propose accepting it for the conference. The underlying idea is simple and compelling, well explained and analyzed with a fresh eye and original techniques.
I think it could be made an even stronger work by including:
- A more detailed account of how the presented results relate to recent theoretical advancements in POMDPs, e.g., under which structural properties on the emission function learning in POMDP with memory traces is nearly optimal;
- A more detailed account of what the study suggests for POMDP practitioners. Practical methods include various approaches for handling long-term dependencies in the history, such as LSTM/GRU architectures (e.g., Hausknecht & Stone 2015 "Deep recurrent Q-learning for POMDPs"), history summarization with VAEs (like Dreamer, Hafner et al 2023 "Mastering diverse domains through world models"), transformers (Parisotto et al 2019 "Stabilizing transformers for reinforcement learning"). How does memory traces findings connect to those? Any rpactical takeaway from the analysis?

An interesting aspect that is not touched in the paper is the following: Memory traces have a continuous hyper-parameter lambda, which seems easier to optimize from data online w.r.t. the size of the window. Especially, estimating the gradient of an "hyper-policy over lambda" looks doable...

Memory traces may have another interesting use other than (general) POMDPs, such as RL with trajectory-dependent objectives: See Hazan et al 2019 "Provably efficient maximum entropy exploration", Chatterji et al "On the theory of reinforcement learning with once-per-episode feedback", Mutti et 2023 "Convex reinforcement learning in finite trials", Kausik et al 2024 "A theoretical framework for partially observed reward-states in RLHF" among others.

**Other Strengths And Weaknesses:**

Strengths
- Simple yet clever method for history summarization in POMDPs.
- Original arguments and techniques;
- Clear presentation of the ideas and compelling analysis.

Weaknesses

While I could not spot any major weakness in the work, my feeling is that more effort could have been made to connect the reported results to the literature of POMDPs (theoretical and empirical).

**Questions For Authors:**

Rewards depend on observations instead of states? Also, why the policy is Markovian and conditioned on true states!?

The paragraph between lines 55-65 (right) is somewhat confusing. The reward is defined as a function from observations to reals, but in POMDP the reward is typically a function from states to reals. Perhaps is a typo there, or the authors mean a function of history of observations? Also, why the policy is a function between states and probability over actions? In POMDPs one typically considers functions from history of observations to actions...

The lines 281-283 read "Thus, there exists no environment where windows outperforms memory traces in general". Are there any structural assumptions on the POMDP for which one or the other is preferable instead?

In spirit, the results are saying: With $\lambda < 1/2$ windows and memory traces are on par. With $\lambda > 1/2$, there exist environments where traces outperform windows. The latter does not exclude the existence of environments where windows are better in the $\lambda > 1/2$ regime, or am I missing some nuances?

**Relation To Broader Scientific Literature:**

To the best of my knowledge, the reported study is original and novel. Most of the (recent) theoretical literature on POMDPs focuses on structural assumptions on the emission functions that makes learning (with a window) statistically tractable. The study of this paper is mostly orthogonal to those results. In terms of methodology, I am not aware of previous works employing this kind of eligibility traces, although I am not familiar with methodological research in POMDPs.

**Theoretical Claims:**

I only checked (briefly) the proofs reported in the main text.

---

> ### Author Rebuttal · Authors · 2025-03-31
>
> Thank you very much for your positive review! Below, we respond to your critical points and questions.
>
> **Expanding the discussion on connections to prior work on POMDPs.**
> We agree that our discussion on related works is relatively sparse. Based on your suggestions, as well as the other reviewers' suggestions, we will expand this section with a discussion on connections to existing empirical work that focuses on using RNNs and transformers for reinforcement learning in POMDPs.
>
> **Why is the reward signal a function of the observations and not of the states?**
> These two descriptions of POMDPs are equivalent. Given a POMDP with state space $\mathcal X$, observation space $\mathcal Y$, set of possible rewards $\mathcal R$, emission probabilities $p(y \mid x)$, and state-reward function $r: \mathcal X \to \mathcal R$, we can consider the equivalent POMDP with observation space $\tilde{\mathcal Y} = \mathcal Y \times \mathcal R$, emission probabilities $p(\tilde y = (y, r) \mid x) = p(y \mid x)[r = r(x)]$, and observation-reward function $\tilde r(\tilde y = (y, r)) = r$.
> We use the observation-reward definition because it emphasizes that rewards are also quantities that are observed by the agent. In this model, state and reward are conditionally independent given the observation. With the state-reward definition this is not the case, so rewards would technically need to be included explicitly in memory as they could also give information about the state. We are open to changing our definition if it is confusing.
>
> **Why is the policy Markovian?**
> If the policy $\pi$ is Markovian, then the combination of POMDP and $\pi$ describe a hidden Markov model with the same state space $\mathcal X$ as the POMDP. If the policy is not Markovian, then this is no longer true. However, this is irrelevant to our paper, and so we will change the definition in the final version as follows. Consider the general setup consisting of a POMDP with state space $\mathcal X$, observation space $\mathcal Y$, action space $\mathcal U$, transition dynamics $p(x_{t + 1}\mid x_t, u_t)$ and emission probabilities $p(y_t\mid x_t)$, and a general agent with agent state space $\mathcal Z$, agent state update probabilities $\pi(z_{t+1}\mid z_t, y_{t+1})$ and policy $\pi(u_t\mid z_t)$. Together, these describe a hidden Markov model with state space $\Psi = \mathcal X \times \mathcal Z$, observation space $\mathcal Y$, transition dynamics
> $$p(\psi_{t+1} = (x_{t + 1}, z_t) \mid \psi_{t} = (x_{t}, z_{t-1})) = \sum_{y_t}p(y_t\mid x_t)\pi(z_t\mid z_{t - 1}, y_t)\sum_{u_t}\pi(u_t\mid x_t)p(x_{t + 1}\mid x_t, u_t),$$
> and emission probabilities
> $$p(y_t\mid \psi_t = (x_t, z_{t-1})) = p(y_t\mid x_t).$$
> Thus, as none of our results depend on the state space of the HMM, this definition shows that our results also apply with non-Markovian policies.
>
> **Are there any structural assumptions on the POMDP for which one or the other is preferable instead?**
> This is an interesting question that we do not yet know the answer to. Our theoretical results suggest that it is always possible to use memory traces and perform at least as well as the best window-based approach. However, this is only true for the optimal choice of $\lambda$, which in general will not be known. We are interested in investigating if there is a way to connect properties of the transition and emission dynamics of the POMDP to the optimal choice for $\lambda$.
>
> **If $\lambda > 1/2$, are there environments where windows outperform memory traces?**
> Yes, there are (e.g., see example A.1). However, there is a "simple fix": just choose a different $\lambda$. If $\lambda$ is set to a value less than $1/2$, Theorem 5.5 guarantees that no window will outperform the memory trace. The T-maze, which we analyze in Theorem 5.8, is qualitatively different: there is no "simple fix" to make a window-based approach efficient here (e.g., by choosing a different $m$). Thus, there is no environment where a window-based approach outperforms memory traces for all $\lambda$. However, there are environments where a memory trace approach outperforms windows for all $m$ (e.g., the T-maze).
>
> **Other suggestions.**
> Thank you very much for your additional suggestions. We are working on a follow-up project right now, and will have a look at your references on RL with trajectory-dependent objectives. We are also examining if it is possible to learn $\lambda$ online.

---

> > ### Comment · Reviewer_hzft · 2025-04-03
> >
> > Dear authors,
> >
> > Thanks for the thorough replies to my comments and additional clarifications. I am even more convinced of my original evaluation: I will advocate for acceptance.
> >
> > Reward definition: No need to change the formulation, but perhaps reporting the equivalence argument in a footnote can avoid potential doubts!

---

### Official Review · Reviewer_iWAi · 2025-03-12

**Overall Recommendation:** 4

**Summary:**

The paper studies learning with memory traces as an alternative to finite length history windows to solve POMDPs. Memory traces are exponential moving average of observations. In offline RL, when the forgetting factor \lambda < 0.5, they are shown to be equivalent to learning with windows in capacity and sample complexity. If lambda is larger, they are significantly more efficient. The paper also shows that the memory traces can be easily incorporated into online RL algorithms and outperforms the windows approach.

**Claims And Evidence:**

The claims are well supported by theorems and proofs. The empirical performance of the memory traces approach is illustrated in Sutton's random walk experiment and in a Minigrid version of T-maze.

**Essential References Not Discussed:**

The paper cites all relevant prior work to the best of my knowledge.

**Ethical Review Concerns:**

================= Post-Rebuttal Update =================

Thank you for the responses to the reviews. I stand by my evaluation and think that the paper should be accepted.

**Experimental Designs Or Analyses:**

The experimental results on the random walk domains and the T-maze domain validate the claims but are relatively small. The authors are encouraged to consider more complex and compelling experiments.

**Methods And Evaluation Criteria:**

The paper's main contributions are theoretical in nature. The key result that the memory traces are injective if lambda is rational is surprising. The sample complexity results are carefully proved using relevant mathematical machinery, e.g., Kirszbraun's theorem.
The experimental results are based on relatively small domains and are proof-of-concept variety. Larger experimental domains would make the paper more attractive.

**Other Comments Or Suggestions:**

The paper is well written in spite of its highly theoretical content.

**Other Strengths And Weaknesses:**

The paper's theoretical claims and proofs make a strong contribution to the theory of POMDPs. The paper also offers empirical evaluations in multiple domains, but these results are preliminary and can be strengthened.

**Questions For Authors:**

It appears that the effectiveness of the method relative to the windows method depends on the value of the forgetting factor. What determines the forgetting factor itself? Under what conditions do we need a forgetting factor > 1/2?

**Relation To Broader Scientific Literature:**

N/A

**Theoretical Claims:**

I have checked the proofs in the main paper but not those in the Appendix. The main proofs appear to be correct.

---

> ### Author Rebuttal · Authors · 2025-03-31
>
> Thank you very much for your positive review! Below, we respond to your critical points and questions.
>
> > Larger experimental domains would make the paper more attractive.
>
> Indeed, the environments we consider in our experiments are more illustrative than realistic. We intend the main contributions of this paper to be theoretical, and we are working on a follow-up project in which we investigate memory traces empirically in more complex environments. The reason for this is that a more thorough empirical evaluation requires a lot more work. For example, the assumption of one-hot observation only makes sense in small tabular POMDPs. A proper treatment of these issues therefore goes beyond the scope of this paper.
>
> > It appears that the effectiveness of the method relative to the windows method depends on the value of the forgetting factor. What determines the forgetting factor itself? Under what conditions do we need a forgetting factor > 1/2?
>
> These are interesting questions that we do not yet know the answers to. All that our theoretical (and empirical) results suggest is that $\lambda$ should usually be larger than $\frac{1}{2}$, as otherwise a window could be used instead. In our follow-up work, we investigate whether it is possible to learn $\lambda$, and if there is a way to connect properties of the transition and emission dynamics of the POMDP to the optimal choice for $\lambda$.

---

### Official Review · Reviewer_s69o · 2025-03-12

**Overall Recommendation:** 3

**Summary:**

The authors propose memory traces as an alternative to fixed-window histories, or “frame stacking,” for addressing partial observability in RL. The concept is loosely related to eligibility traces, amounting to an exponential moving average of the observation stream, which is then fed to the agent as input for the value function and/or policy. The authors prove conditions on $\lambda$ for injective-ness (when the trace is a sufficient statistic for the history), separability, and concentration. Analysis for offline on-policy sample complexity (in terms of minimum value error) is also given, including theorems that relate memory traces to fixed-length windows, and vice versa. The T-Maze environment is given as an instance where memory traces can be proved to be more efficient than windows. Finally, traces and windows are compared in a partially observable random walk, and again in the T-Maze environment using PPO (a deep RL agent). In these environments, memory traces are shown to achieve lower value error and higher average reward, respectively.

**Score raised from 2 to 3 after rebuttal**

**Claims And Evidence:**

Claims are generally supported with convincing evidence.

**Essential References Not Discussed:**

Deep RL using RNNs or attention, e.g.,
- DRQN: https://arxiv.org/abs/1507.06527
- DTQN: https://arxiv.org/abs/2206.01078

**Experimental Designs Or Analyses:**

- Random walk
  - The comparison is unfair here. If I am understanding correctly, the authors’ method uses linear function approximation to evaluate the memory trace, but the fixed-window baseline uses the (integer) history to index into a table. Clearly, the approximation method will perform better due to generalization. It would be far better to produce a window history by concatenating $m$ one-hot vectors; that way, both methods can use linear function approximation. Without this, I am skeptical of the results.
  - The step-size selection is not explained in sufficient detail. It is not reported what step sizes were used to generate Figure 4. The criteria for “best” step size is never defined.
- Deep RL
  - For some reason, two memory traces are given to the authors’ method instead of one. One trace has $\lambda=(k-1)/k$ and the other has $\lambda=0$. This seems unfair for two reasons: ~it has a dynamic $\lambda$ and is not a true exponential average (as I discussed above) and~ it seems to give the agent an extra hint about the current observation. I cannot figure out why this was done other than to make the authors’ performance look better than reality.
  - The PPO optimizer is not discussed. It would not be possible to reproduce the results in the paper from their description.
  - Figure 5 says “average success rate” in the caption but the y-axis label says “average total reward.” I am not sure what the real metric is supposed to be.

**Methods And Evaluation Criteria:**

Overall, the methods and evaluation criteria are pretty good, aside from some experiment design concerns I have in **Experimental Designs or Analyses**.

I think it would be beneficial to test in environments that have more partial observability, as T-maze requires remembering essentially 1 bit of information over just a few time steps.

I also feel that some important baselines are missing, such as RNNs or attention. It is already well-known that a fixed-window strategy is the most naive approach for addressing POMDPs and that it does not scale well. Thus, strictly outperforming a window when there are more sophisticated techniques widely used does not feel sufficient to me.

**Other Comments Or Suggestions:**

- Typo in TLDR of paper: “We analyze the effectiveness of eligibility traces when used as memory a memory in POMDPs.”
- Viewing Figure 6 causes my computer to lag significantly, for some reason.
- Y-axis in Figure 5 is misleading, since it doesn’t start at 0. It exaggerates the difference between methods.
- Typo: “A large $\lambda$ can then reduces” (line 358).
- Proof of theorem 5.8 is only partial; it should be called a sketch.
- Several parts of the paper use “cf.” to refer to a figure or a lemma, but “see” would be more accurate in these instances.
- The proof of lemma 5.4 doesn’t seem necessary in the main paper. It takes up an entire page, but doesn’t appear to be a central result to the paper’s story. The space could be used to better explain the experiments in Section 5, for instance.
- Notation $VE_\mathcal{E}$, where $\mathcal{E}$ means environment, seems incomplete. Doesn’t the minimum-achievable VE also depend on the policy, $\pi$?
- Why is the reward function defined in terms of observations instead of states? This doesn’t seem to match standard POMDP formulations, which define the reward based on the underlying state.

**Other Strengths And Weaknesses:**

**Strengths**
- The injective nature of memory traces (histories can usually be uniquely recovered) is surprising and interesting. I like the visualization of the trace space as a Sierpinkski triangle (Figure 2), which helped convince me that this is true.
- The $\lambda$/$m$ sweep in Figure 4 is a nice comparison of the two different algorithms (traces vs windows) across two different hyperparameters, especially with how $m$ and $\lambda$ are lined up. Although the authors should note how they are doing this (it seems they are using the relation $m = 1/(1-\lambda)$, but it should be explained in the paper).

**Weaknesses**
- One limitation is that, despite the authors’ proof that memory traces are usually injective, the exponential decay of the traces will squash most useful information after more than a few time steps. This can be seen clearly in Figure 2, where even in a low-dimensional setting with just $\lambda = 0.7$, the points become very hard to distinguish. It is not clear to me that a function approximator would be able to meaningfully disentangle these histories without extreme overfitting.
- It would be nice to see a comparison on harder, more partially observable environments. Would this scale, e.g., with image-based observations? The exponential average would smear moving or appearing/vanishing objects across the frame, and it is not clear to me if this would be useful or distracting to the agent.
- I am confused by the decision to define sample efficiency in terms of the minimum-achievable value error, $\min \bar{VE}_\mathcal{E}$. Just because this value is lower does not mean that learning requires fewer samples, does it? For example, intuitively, I would think that a linear function does not have a low VE but would be easier to learn because it is convex.

**Questions For Authors:**

- Does the lack of injective guarantees for infinite memory traces limit the applicability of memory traces to continuing (i.e., non-episodic) environments?
- ~“Thus, there exists no environment where windows outperform memory traces in general.” But is the reverse not true, i.e., traces cannot outperform windows, either? Consider a fully observable POMDP where $O_t = S_t$ always. Then wouldn’t a fixed window of $m=1$ be much easier to learn than the exponential average, which smears irrelevant information across time steps?~

**Relation To Broader Scientific Literature:**

There is a short related work section in which the authors cite three papers which theoretically analyzed windows of fixed length. It does not appear that any previous works studied exponential windows. Some deep RL papers which use windows as input are mentioned. There is no discussion of more sophisticated memory techniques, as far as I see.

A short discussion regarding the inspiration from eligibility traces is also included. However, the relationship to eligibility traces is merely coincidental, in my opinion, and is overemphasized in the paper. TD($\lambda$) is technically not an exponential average; the weights do not sum to $1$ and nothing is being averaged. The approaches are orthogonal; it would be possible to have an eligibility trace in which the feature vectors are obtained using the memory trace proposed by the authors.

**Theoretical Claims:**

The theoretical claims are correct, to my knowledge.

~However, the proof of Theorem 5.8 is surprising, as it prescribes setting $\lambda = (k-1)/k$ in the T-maze. In this case, the exponential average degenerates into an unweighted average of the past $k$ observations, sacrificing the injective guarantees of Theorem 4.1. Does this suggest that truly exponential traces are actually suboptimal in this environment? It seems to contradict the main message of the paper.~

---

> ### Author Rebuttal · Authors · 2025-03-31
>
> Thank you very much for your thorough review of our paper! Below, we respond to your critical points and questions. Due to limited space, we cannot respond to every point, but we will do our best to incorporate all your suggestions in the final version.
>
> **Proof of Theorem 5.8 and the choice of $\lambda = \frac{k - 1}{k}$ in the T-maze.**
> The full proof of Theorem 5.8 is in Appendix B (p. 14). In this proof, as well as in our PPO experiment, we fix the forgetting factor to $\lambda = \frac{k - 1}{k}$, where $k$ is the corridor length of the T-maze. Importantly, this corridor length is fixed, and does not change during an episode; it is not to be confused with the time step $t$. Thus, $\lambda$ is **not** dynamic, and the memory trace is indeed an exponentially weighted average.
>
> **Concerns about the PPO (T-maze) experiment**
> - In the PPO experiment, we additionally give the memory trace agent access to the current observation (or, equivalently, we use a second memory trace with $\lambda = 0$). We did not do this to give an edge over the frame stacking agent. Indeed, the frame stacking agent also has access to the current observation. Instead, it was a heuristic choice and a demonstration that a practitioner is not constrained to using just one memory trace. We are happy to run additional experiments with only a single memory trace.
> - We use the Adam optimizer with a linear learning rate decay from $0.0003$ to $0$ as suggested by [Huang et al. 2022].
> - In the T-maze minigrid, the agent receives a reward of $1$ if it succeeds, and a reward of $0$ if it does not (and $0$ in all nonterminal transitions). Thus, the "average success rate" is the same as the "average total reward". A random agent would achieve an average success rate of $0.5$. The observed performance slightly below $0.5$ is due to episodes exceeding the maximum episode length of $5(k + 2)^2$ steps in a T-maze with corridor length $k$. We will clarify this in the article. Does this address your concerns about the Y-axis in Figure 5?
>
> **Concerns about the TD (random walk) experiment**
> - Thank you for your feedback. In this experiment, our goal was to compare memory traces to the full window approach whose sample efficiency we analyze in Sec. 5. However, we agree that we could instead concatenate observations to improve sample efficiency, as we do in the PPO experiments. While we are not aware of this approach in the context of linear function approximation, we ran the experiments that you suggested, please see [here](https://anonymous.4open.science/api/repo/icml2025-rebuttal-trace/file/sutton-concat.pdf). All three algorithms ran for 100,000 steps (1/100th of Fig. 4) with a grid search over step sizes. We see that concatenation makes it possible to learn with very long windows, and that this approach almost reaches the best value error of memory traces. However, it requires almost 20 times the memory to achieve this, and it lacks the theoretical guarantees that we develop in Sec. 5.
> - The step size used in the original TD experiments (Fig. 4) for the window approach is $\alpha = 0.0001, 0.0002, 0.005, 0.05, 0.2, 0.5$ for $m = 1, 2, 3, 4, 5, 6$, respectively. These values were chosen by a grid search optimizing the average value error. For the memory trace, the step size was kept constant at $\alpha = 0.0002$ for all values of $\lambda$. This value was also determined by a grid search.
>
> **Further points**
> - We will expand Appendix C in the final paper. (optimizer, step sizes, ..)
> - Essential References. Thank you for these suggestions. We will include them in the final version.
> - Relation to eligibility traces. We do believe that there is a close connection to eligibility traces, and will include an extended discussion in the final version.
> - Definition of sample efficiency. A low minimum value error does not correspond to low sample efficiency. Could you please explain what gave this impression? We use Theorem 5.1 to tie sample efficiency to the metric entropy.
> - Definition of environment $\mathcal E$. In our definition of $\mathcal E$, the policy is assumed fixed (line 107).
> - More realistic experiments & observation-reward function. Please see our response to Reviewer tJLm.
> - Q1: Lack of injectivity guarantee. Could you please elaborate on this question? We provide one experiment in a continuing environment and one in an episodic environment.
> - Q2: "Traces cannot outperform windows". In a fully observable POMDP, we can use $\lambda = 0$ to achieve the same result. Theorem 5.5 guarantees that, if learning with windows is tractable, then learning with traces is also tractable.
>
> Please let us know if you still have any concerns, or if our responses were not satisfactory. If you agree that we have addressed your questions and provided the necessary clarifications, we kindly request raising our paper’s score based on the responses provided. We are happy to incorporate further feedback into the paper to improve clarity and remove ambiguity.

---

> > ### Comment · Reviewer_s69o · 2025-04-04
> >
> > Thanks for the rebuttal. I have crossed out any questions/concerns which are no longer applicable. I also follow up some more points below:
> >
> > ---
> >
> > > The full proof of Theorem 5.8 is in Appendix B (p. 14).
> >
> > Thanks for the clarification about the static $\lambda$. I see that the full proof is in the appendix, but my point was that the proof written in the main paper is abbreviated and should be labeled as a sketch to avoid confusion. Typically, this is done by writing *Proof (sketch).*
> >
> > ---
> >
> > > We are happy to run additional experiments with only a single memory trace.
> >
> > Please do if it is possible. I feel this is crucial to remain true to the theory and to fully connect the theory with the experiment results. Using two memory traces instead of one has all sorts of theoretical consequences not addressed by the paper (e.g., strengthening injectivity and information content) and it would be more appropriate for future work that analyzes dual traces, in my opinion.
> >
> > ---
> >
> > > Does this address your concerns about the Y-axis in Figure 5?
> >
> > Yes! Thank you, but please make sure to clarify this in the paper.
> >
> > ---
> >
> > > We see that concatenation makes it possible to learn with very long windows, and that this approach almost reaches the best value error of memory traces. However, it requires almost 20 times the memory to achieve this, and it lacks the theoretical guarantees that we develop in Sec. 5.
> >
> > ---
> >
> > The concatenation of observations is the standard approach for windows using function approximation, being exactly analogous to frame stacking with images and convolutional networks. It does greatly increase memory consumption and computation, but these are deficiencies of the window approach---which is is exactly why I wanted the experiment to be included. It demonstrates the benefits of memory traces while eliminating generalization as a confounder.
> >
> > > Relation to eligibility traces. We do believe that there is a close connection to eligibility traces, and will include an extended discussion in the final version.
> >
> > ---
> >
> > Mathematically, it is clear why memory traces look like eligibility traces, but I personally would refrain from calling it a close connection. (If you have a counterargument, though, I'd love to hear it.) Memory traces learn a value function over the trace itself, i.e., if $z_t = (1-\lambda) (x_t + \lambda x_{t-1} + \dots)$ is the memory trace, then it learns $V(z_t)$. Eligibility traces learn a value function over the original inputs, $V(x_t)$, using the reinforcement signal $\delta_t z_t$. The only similarity is the computation of the traces $z_t$, but their roles are completely different and orthogonal. For example, to learn $V(z_t)$ using eligibility traces, you would need an eligibility trace over memory traces: $z^\prime_t = (1-\lambda) (z_t + \lambda z_{t-1} + \dots)$ and then reinforce according to $\delta_t z^\prime_t$. One is changing the input space of the value function, the other is changing the reinforcement signal applied to each input and, as a consequence, the fixed point of the TD update.
> >
> > ---
> >
> > > Definition of sample efficiency. A low minimum value error does not correspond to low sample efficiency. Could you please explain what gave this impression? We use Theorem 5.1 to tie sample efficiency to the metric entropy.
> >
> > I was confused by the statement immediately after Theorem 5.1: "While this result does not guarantee that function classes with large metric entropy are less suitable for learning, it suggests that a good value function is more easily learned if both $VE(\mathcal{F})$ and $H_\epsilon(\mathcal{F})$ are small." Given that the start of section 5 said that the analysis would focus on sample complexity, the phrase "more easily learned" seemed to suggest that both a small value error and a small metric entropy would improve sample complexity/efficiency.
> >
> > ---
> >
> > > Q1: Lack of injectivity guarantee. Could you please elaborate on this question? We provide one experiment in a continuing environment and one in an episodic environment.
> >
> > Theorem 4.1 says that finite histories are needed for injectivity. This implies infinite histories (continuing environments) are not injective, but the paper doesn't discuss the ramifications of this. I was curious if it would limit the effectiveness of memory traces in non-episodic environments.

---

> > > ### Author Response · Authors · 2025-04-07
> > >
> > > Thank you again for your time in reviewing and responding!
> > >
> > > **New PPO experiments.**
> > > We ran the additional experiments that you requested: PPO in the T-maze with only a single memory trace (with the same forgetting factor $\lambda_k \doteq \frac{k - 1}{k}$ as before). Please see the results [here](https://anonymous.4open.science/api/repo/icml2025-rebuttal-trace/file/ppo2.pdf). We have also included a purely reactive agent as a baseline which should help put the other performances into perspective. It can be seen that a single memory trace is still preferable to frame stacking in long corridors, and that adding a second one ($\lambda = 0$) helps even more. For very long corridors, we see that the reactive agent struggles to even find the end of the corridor reliably before the time runs out.
> > >
> > > **Relation to eligibility traces.**
> > > The reason why we mention eligibility traces is that this is where the idea for memory traces stems from. Eligibility traces serve as a "memory" for the TD algorithm, where they are used for temporal credit assignment. However, you are completely right that it is possible to use memory traces and eligibility traces in parallel, and that they are designed for different purposes.
> > >
> > > We believe that there is a connection, because our empirical results (Fig. 4) show that TD(0) algorithm, when used with memory traces ($\lambda$ close to $1$), converges very close to the optimal memory trace solution. This is in contrast to the window/memoryless value functions, where the TD(0) fixed point is relatively far from the optimal solution. While memory traces indeed change the input space of the value estimate, we can still compare learned weight vectors for different $\lambda$. For example, given a linear (memory trace) value estimate $\hat v(z) = w^\top z$, we can define a state value estimate as $\tilde v(x) \doteq \mathbb E [w^\top z_t \mid x_t = x] = \tilde w^\top x$, with $\tilde w = (I - \lambda \hat T)^{-\top}E^\top w$, where $\hat T$ is the time-reversed transition kernel of the HMM, and $E$ is the emission matrix. Note that in the fully observed case ($E = I$), this expression reduces to the original weight vector if $\lambda = 0$. Thus, $\lambda$ indeed affects the convergence point, although it is in a different way than TD($\lambda$).
> > >
> > > As this connection is still somewhat unclear, we agree that it is best not to include it in this paper, and we will make sure to make the distinction between the two concepts as clear as possible.
> > >
> > > **Definition of sample efficiency.**
> > > We see why this sentence may be confusing, and we will change the phrasing in the final paper. The minimum achievable value error in $\mathcal F$ only plays a role in that, if it is very large, then it is impossible to learn a "good" value function in $\mathcal F$ (as none exists). However, it is unrelated to the sample complexity of learning.
> > >
> > > **Injectivity.**
> > > Thank you for clarifying. Injectivity may sound desirable at first, as it means that no information is lost, but this actually makes learning harder (it is equivalent to keeping the whole history). Injectivity still holds for infinite histories when $\lambda < 1/2$ (line 204, left column). It is for this reason that we analyze Lipschitz continuous functions of memory traces rather than general functions. The Lipschitz constant limits the "resolution" of the functions that we consider, and Theorem 5.4 shows that the metric entropy (and thus, the complexity of learning) increases as we allow larger Lipschitz constants $L$. Without limiting $L$ (or introducing some other constraint), we could therefore not learn efficiently (it would be equivalent to learning with complete, infinite-length histories).
> > >
> > > Thank you very much for showing interest in our work! We will make sure to address your remaining points in the final paper.

---

### Official Review · Reviewer_tJLm · 2025-03-13

**Overall Recommendation:** 3

**Summary:**

The authors propose a novel method for handling observations in reinforcement learning for partially observable systems.
Their method represents the history of observations with an exponential moving average. The authors analyze sample complexity bounds for offline on policy evaluation. The novel method is compared with the traditionally used moving windows both theoretically and empirically.

**Claims And Evidence:**

All claims are supported by evidence.

**Essential References Not Discussed:**

I think it would have been interesting to discuss the use of RNNs and transformers to represent the history of observations.

**Experimental Designs Or Analyses:**

The experiment contains classical benchmark tasks and compares to the traditional method of moving windows.
The experiments for value evaluation and for control are sound.

**Methods And Evaluation Criteria:**

The authors use classical benchmark tasks that are well suited for the problem at hand.

**Other Comments Or Suggestions:**

It would be interesting to discuss how to chose $\lambda$ in different environments.

**Other Strengths And Weaknesses:**

Strength:
The authors effectively introduce the concept of memory traces with a well-designed example in Figure 1. This simple yet insightful illustration (using the T-maze environment) clearly demonstrates the motivation behind memory traces and their advantage over traditional window-based approaches.

Weakness:
The evaluation is on relatively simple tasks. Exploring a broader range of environments could further establish the generality of the approach.

**Questions For Authors:**

Do you assume the agent remembers its actions?  Actions also provide important context for the agent.

I am more familiar with different descriptions of the POMDP setting, where the reward is a mapping from the latent state to the reward. Does this impact the proposed method?

**Relation To Broader Scientific Literature:**

This paper is a theoretical contribution to the POMDP literature. While most current methods use neural network based approaches like RNNs or transformers, they lack theory. This paper contributes to the understanding of observation history representation in POMDPs.

**Theoretical Claims:**

I did not independently verify all the proofs in detail.

---

> ### Author Rebuttal · Authors · 2025-03-31
>
> Thank you very much for your positive review! Below, we respond to your critical points and questions.
>
> > I think it would have been interesting to discuss the use of RNNs and transformers to represent the history of observations.
>
> We are happy to include a brief discussion of the connections to RNN and transformer-based methods. Indeed, it is possible to view memory traces as a kind of simplified RNN, while the window-based approach has a fixed context length like transformers. Based on the other reviews, we will include references to https://arxiv.org/abs/1507.06527 and https://arxiv.org/abs/2206.01078.
>
> > The evaluation is on relatively simple tasks. Exploring a broader range of environments could further establish the generality of the approach.
>
> Indeed, the environments we consider in our experiments are more illustrative than realistic. We intend the main contributions of this paper to be theoretical, and we are working on a follow-up project in which we investigate memory traces empirically in more complex environments. The reason for this is that a more thorough empirical evaluation requires a lot more work. For example, the assumption of one-hot observation only makes sense in small tabular POMDPs. A proper treatment of these issues therefore goeses beyond the scope of this paper.
>
> > It would be interesting to discuss how to choose $\lambda$ in different environments.
>
> We agree that this is an interesting question that is not fully addressed in the paper. All that our theoretical (and empirical) results suggest is that $\lambda$ should usually be larger than $\frac{1}{2}$, as otherwise a window could be used instead. In our follow-up work, we investigate whether it is possible to learn $\lambda$, and if there is a way to connect properties of the transition and emission dynamics of the POMDP to the optimal choice for $\lambda$.
>
> > Do you assume the agent remembers its actions? Actions also provide important context for the agent.
>
> In this paper, we define memory traces only based on the observations. However, it is easy to include actions in the traces as well by regarding them as observations, i.e., by considering the augmented observation space $\tilde{\mathcal Y} = \mathcal Y \times \mathcal U$.
>
> > I am more familiar with different descriptions of the POMDP setting, where the reward is a mapping from the latent state to the reward. Does this impact the proposed method?
>
> These two descriptions of POMDPs are equivalent, so it would not impact the proposed method. Given a POMDP with state space $\mathcal X$, observation space $\mathcal Y$, set of possible rewards $\mathcal R$, emission probabilities $p(y \mid x)$, and state-reward function $r: \mathcal X \to \mathcal R$, we can consider the equivalent POMDP with observation space $\tilde{\mathcal Y} = \mathcal Y \times \mathcal R$, emission probabilities $p(\tilde y = (y, r) \mid x) = p(y \mid x)[r = r(x)]$, and observation-reward function $\tilde r(\tilde y = (y, r)) = r$.
> We use the observation-reward definition because it emphasizes that rewards are also quantities that are observed by the agent. In this model, state and reward are conditionally independent given the observation. With the state-reward definition this is not the case, so rewards would technically need to be included explicitly in memory as they could also give information about the state. We are open to changing our definition if it is confusing.

---

### Decision · Program_Chairs · 2025-05-01

**Decision:**

Accept (poster)

**Comment:**

Thanks to some useful back and forth between the reviews and the authors, there is convergence on a positive recommendation for this paper based upon its novel use of an eligibility-trace like concept for memory. The only lingering disappointment is that more ambitious experiments are relegated to future work. This is somewhat relatable given the resources required to do thorough experimentation, but also somewhat unsatisfying.